# A prediction model of working memory across health and psychiatric disease using whole-brain functional connectivity

Masahiro Yamashita[1], Yujiro Yoshihara[2], Ryuichiro Hashimoto[3], Noriaki Yahata[4,5], Naho Ichikawa[6], Yuki Sakai[1,7], Takashi Yamada[1,3], Noriko Matsukawa[2], Go Okada[6], Saori C Tanaka[1], Kiyoto Kasai[4,8], Nobumasa Kato[3], Yasumasa Okamoto[6], Ben Seymour[1,9,10]*, Hidehiko Takahashi[2]*, Mitsuo Kawato[1]*, Hiroshi Imamizu[1,11]*

[1]Brain Information Communication Research Laboratory Group, Advanced Telecommunications Research Institute International, Kyoto, Japan; [2]Department of Psychiatry, Graduate School of Medicine, Kyoto University, Kyoto, Japan; [3]Medical Institute of Developmental Disabilities Research, Showa University, Tokyo, Japan; [4]Department of Youth Mental Health, Graduate School of Medicine, The University of Tokyo, Tokyo, Japan; [5]Molecular Imaging Center, National Institute of Radiological Sciences, Chiba, Japan; [6]Department of Psychiatry and Neurosciences, Graduate School of Biomedical and Health Sciences, Hiroshima University, Hiroshima, Japan; [7]Department of Psychiatry, Graduate School of Medical Science, Kyoto Prefectural University of Medicine, Kyoto, Japan; [8]Department of Neuropsychiatry, Graduate School of Medicine, The University of Tokyo, Tokyo, Japan; [9]Computational and Biological Learning Laboratory, Department of Engineering, University of Cambridge, Cambridge, United Kingdom; [10]Center for Information and Neural Networks, National Institute of Information and Communications Technology, Osaka, Japan; [11]Department of Psychology, The University of Tokyo, Tokyo, Japan

*For correspondence:
bjs49@cam.ac.uk (BS);
hidehiko@kuhp.kyoto-u.ac.jp (HT);
kawato@atr.jp (MK);
imamizu@gmail.com (HI)

**Competing interests:** The authors declare that no competing interests exist.

**Abstract** Working memory deficits are present in many neuropsychiatric diseases with diagnosis-related severity. However, it is unknown whether this common behavioral abnormality is a continuum explained by a neural mechanism shared across diseases or a set of discrete dysfunctions. Here, we performed predictive modeling to examine working memory ability (WMA) as a function of normative whole-brain connectivity across psychiatric diseases. We built a quantitative model for letter three-back task performance in healthy participants, using resting state functional magnetic resonance imaging (rs-fMRI). This normative model was applied to independent participants (N = 965) including four psychiatric diagnoses. Individual's predicted WMA significantly correlated with a measured WMA in both healthy population and schizophrenia. Our predicted effect size estimates on WMA impairment were comparable to previous meta-analysis results. These results suggest a general association between brain connectivity and working memory ability applicable commonly to health and psychiatric diseases.
DOI: https://doi.org/10.7554/eLife.38844.001

## Introduction

Working memory is a goal-directed active information maintenance and manipulation in mind, forming a foundation for diverse complex cognitive functions, learning, and emotion regulation

(*Baddeley, 2003*; *Cowan, 2014*; *Etkin et al., 2015*; *Otto et al., 2013*). A range of psychiatric disorders commonly shows working memory deficits, although the severity of the deficits is dependent of psychiatric diagnosis (*Forbes et al., 2009*; *Lever et al., 2015*; *Millan et al., 2012*; *Snyder, 2014*; *Snyder et al., 2015*). Working memory emerges by coordinating multiple related processes from sensory perception, cognitive control (e.g. updating, focused attention), to motor action and thus requires close communication among widespread brain regions (*D'Esposito and Postle, 2015*; *Eriksson et al., 2015*; *Nee et al., 2013*; *Owen et al., 2005*; *Postle, 2006*; *Rottschy et al., 2012*).

Functional connectivity (FC) quantifies how brain regions are temporally coordinated and is increasingly used to examine brain network architecture. Resting state (i.e. task-free) FC has been associated with a wide range of individual traits (*Baldassarre et al., 2012*; *Dosenbach et al., 2010*; *Lewis et al., 2009*; *Seeley et al., 2007*). For example, whole-brain FC models have recently demonstrated that sets of functional connections across widespread brain regions can predict performance on cognitive tasks (*Finn et al., 2015*; *Rosenberg et al., 2016*; *Smith et al., 2015*; *Yamashita et al., 2015*). These findings suggest that specific cognitive processes (e.g. memory, attention) may be represented by the corresponding interaction patterns among distributed brain networks, at least among healthy populations.

Functional connectivity has also provided insight into the biological basis of psychiatric disorders and shown that different diagnoses are related to unique patterns of FC (*Baker et al., 2014*; *Harrison et al., 2009*; *Kaiser et al., 2015*; *Yahata et al., 2016*). For example, a whole-brain FC-based model has been shown to reliably predict autism spectrum disorder (ASD), as well as individual clinical scores (*Emerson et al., 2017*; *Lake et al., 2018*; *Yahata et al., 2016*), suggesting that FC disruption is quantitatively relevant to behavioral abnormality. More broadly, this suggests that a specific relationship between FC and behavior might exist across many disparate diagnoses that have common symptoms, such as impairments in working memory.

With the above issues in mind, we set out to examine competing hypotheses about the relationship between FC and working memory ability (WMA) across healthy populations and a range of psychiatric diagnoses. In this study, we define working memory ability simply as a summary index of general working memory performance, without specializing sensory modality and underlying subfunctions. The first hypothesis proposes a distinct FC-WMA relationship for each diagnosis, rationalized by the fact that each psychiatric diagnosis is characterized by differential alterations in FC (*Baker et al., 2014*; *Harrison et al., 2009*; *Kaiser et al., 2015*; *Yahata et al., 2016*). This hypothesis predicts that the FC-WMA relationship among healthy populations will fail to generalize in predicting impairments across different diagnoses. The alternative hypothesis proposes a common FC-WMA relationship across health and multiple diagnoses. The rationale for this hypothesis is that previous studies have suggested that several cognitive functions, such as attention and memory, generalize to predict behavior in patients as well as in healthy populations (e.g. *Kessler et al., 2016*; *Lin et al., 2018*; *O'Halloran et al., 2018*; *Rosenberg et al., 2016*). This hypothesis predicts that a FC-WMA relationship estimated from whole-brain functional connections in healthy populations will generalize to predict working memory impairment across diagnoses.

To test these hypotheses, we built a prediction model of working memory ability in a letter 3-back task using whole-brain FC among a healthy population. Then, we examined whether the model was predictive of individual differences in behaviorally measured working memory ability not only in healthy individuals but also in individuals with schizophrenia. Moreover, we examined whether the model was predictive of group differences in working memory ability among four different psychiatric diagnoses by comparing the predicted effect sizes across diagnoses.

## Results

### Design

We constructed a prediction model of working memory ability among healthy individuals recruited at ATR (Advanced Telecommunications Research Institute International), Japan (ATR dataset; *Figure 1A*). To test its generalizability, independently collected resting state fMRI (rs-fMRI) was entered into this model to predict individual working memory ability. Specifically, we applied the model to independent test datasets of healthy individuals in the USA (the Human Connectome Project dataset, HCP dataset) and schizophrenia patients and their controls (*Figure 1B*). The predicted

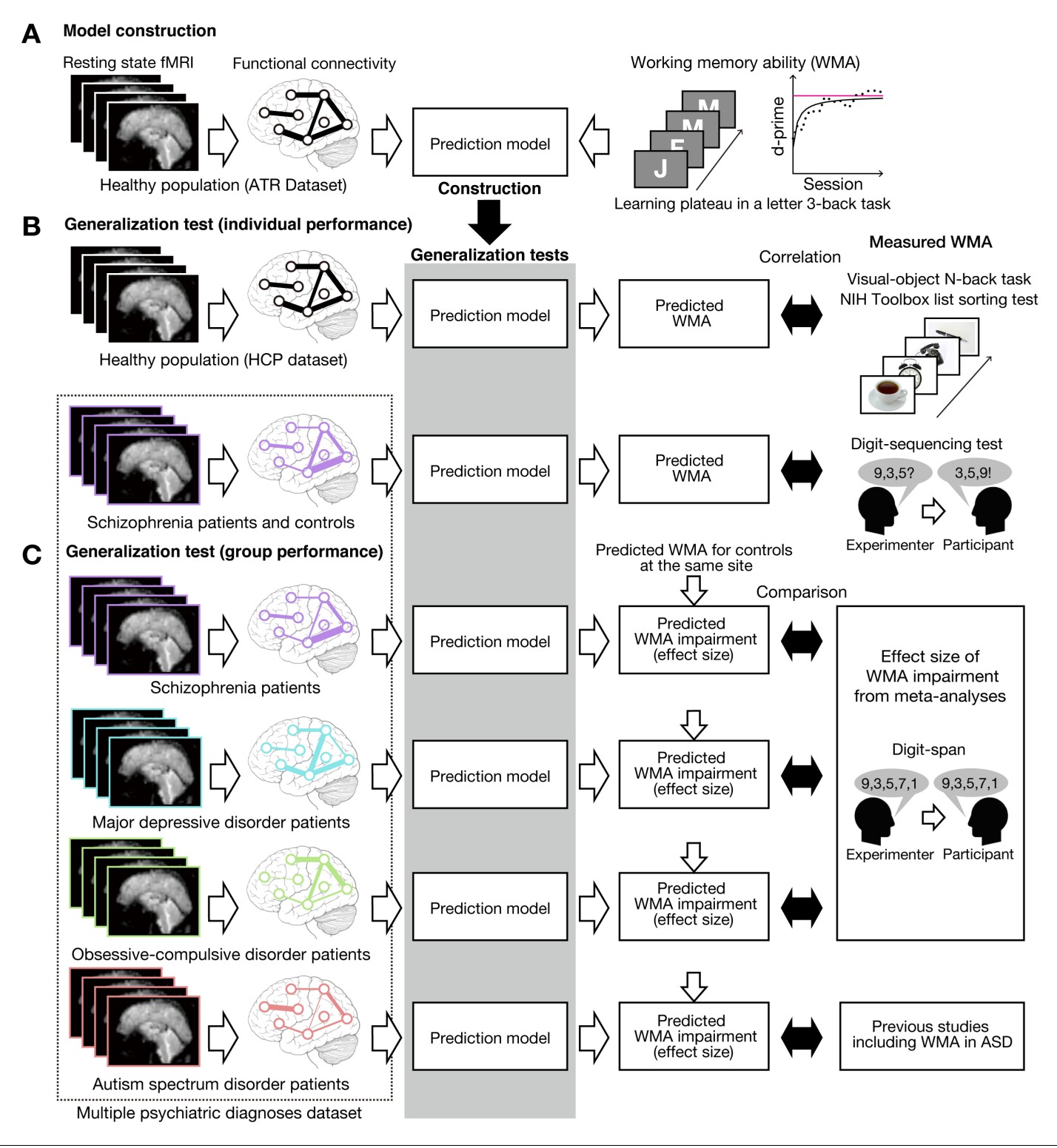

**Figure 1.** Schematic diagram of model construction and generalization tests using independent datasets. (**A**) Model was developed using a whole-brain resting state FC and a learning plateau of a letter 3-back task within healthy individuals from ATR dataset. (**B**) We applied the model to resting state FC patterns and predicted individual participant's working memory ability. We first examined the external validity using an independent USA healthy dataset (HCP dataset: the upper flow chart in (**B**)). The predicted working memory ability was compared to actual working memory performance (visual-object N-back task and the NIH toolbox list sorting test). Then we examined the generalizability to a clinical population using a schizophrenia dataset (the lower flow chart in (**B**)). The predicted working memory ability was compared to actual working memory score measured by Digit

*Figure 1 continued on next page*

*Figure 1 continued*

sequencing test. (C) Using the multiple psychiatric diagnoses dataset, degree of working memory impairment for each diagnosis was predicted as differences from corresponding controls. The predicted impairments were validated by previous meta-analysis studies on digit-span across multiple diagnoses. Note that the HCP dataset's task stimuli images are just illustration purpose and different from the original stimuli.

DOI: https://doi.org/10.7554/eLife.38844.002

working memory ability was compared with actually measured working memory score. We emphasize that these individual differences analyses were performed to examine the relative (i.e. the ability to differentiate between good and bad performers), not absolute accuracy of the prediction model (i.e. the ability to predict specific level of performance). Moreover, the model was applied to patients with psychiatric diagnoses including schizophrenia (SCZ), major depressive disorder (MDD), obsessive-compulsive disorder (OCD), and ASD and also their age- and gender-matched healthy/typically developed controls (multiple psychiatric diagnoses dataset; *Figure 1C*). The effect size estimates of predicted working memory impairments were compared with that of behaviorally observed ones reported in previous meta-analysis studies. We note that individual behavioral scores on working memory ability were available in schizophrenia cohort but not available in other psychiatric diagnoses, and that the same SCZ dataset was analyzed from an individual difference perspective as well as a group-level difference perspective (see below).

## Building prediction model

We used the ATR dataset (*N* = 17, age 19–24 years old) to develop a prediction model of working memory ability. The participants performed a letter 3-back task for about 25 sessions (80–90 min). Working memory ability was quantified by estimating a learning plateau in this task as follows. An individual learning curve was obtained by calculating d-prime for each session, and by smoothing the data points with five-session moving average (*Figure 2—figure supplement 1*). The individual learning curve was fitted by an inverse curve ($y = a - b/x$), where $y$ is a d-prime in the $x$-th session, while $a$ and $b$ is a parameter for learning plateau and learning speed, respectively. We collected resting state fMRI data from each participant, and estimated whole-brain functional connectivity. We used network-level rather than node-level connectivity features to avoid overfitting to training samples (the curse of dimensionality). Specifically, we calculated FC values, based on 18 whole-brain intrinsic networks of BrainMap ICA (*Laird et al., 2011*), for pair-wise between-network (18 × 17/2 = 153) connections and within-network connections.

To evaluate test-retest reliability of this functional connectivity estimation method, we calculated intra-class correlation (ICC) using three external datasets: Beijing Normal University (BNU 1), Institute of Automation, Chinese Academy of Sciences (IACAS 1), and University of Utah (Utah 1). As a result, we obtained ICC values 0.34 ± 0.12 (range 0 to 0.65) for BNU 1, 0.26 ± 0.18 (range 0 to 0.66) for IACAS 1, and 0.21 ± 0.17 (range 0 to 0.59) for Utah one datasets. We found ICC values for the left FPN/right FPN: 0.28/0.23, 0.49/0.47, and 0.11/0.03 for BNU 1, IACAS 1, and Utah one dataset, respectively. According to an interpretation criteria of ICC (*Landis and Koch, 1977*), our connectivity estimation methods yielded 'fair' reliability (0.2 < ICC ≤ 0.4) for the three datasets. These results suggest that test-retest reliability of our methods are comparable to other common connectivity estimation methods (*Birn et al., 2013*; *Noble et al., 2017*).

Using sparse linear regression, individual letter 3-back learning plateaus were modeled as a linear weighted summation of automatically selected 16 functional connectivity values among 15 intrinsic networks (*Figure 2*). The letter 3-back learning plateaus were positively correlated with three functional connectivity values (P1-P3) and negatively correlated with the remaining 13 connectivity values (N1-N13). A contribution ratio of each connection to the working memory ability, which is determined by the product (weight x FC-value) at the connection, is represented as thickness of connection lines in *Figure 2*. *Table 1* describes networks connected by the 16 connections, and the contribution ratio of each connection. The anatomical regions in the network are summarized in *Supplementary file 2*. We did not find a significant correlation between the predicted letter 3-back learning plateau and age ($r = 0.21$, $p = 0.42$), gender ($r = 0.28$, $p = 0.28$), or head motion ($r = - 0.37$, $p = 0.14$). This provided a normative prediction model based on healthy young Japanese participants.

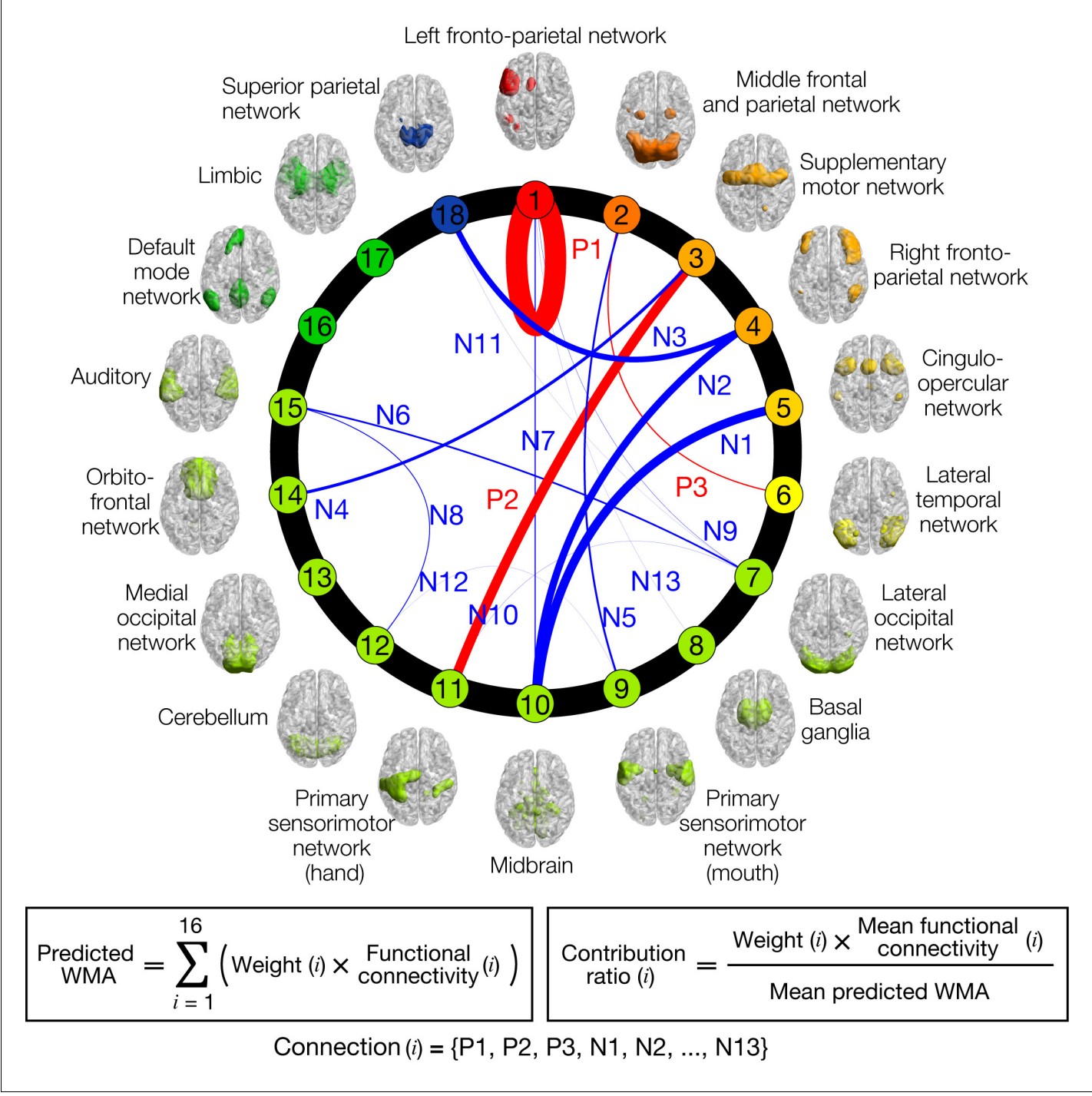

**Figure 2.** Normative model of working memory ability (WMA). Circle plot of networks and their connections in the model. Individual letter 3-back learning plateaus are predicted by a linear weighted summation of 16 FC values at 16 connections selected by a sparse linear regression algorithm. Connection thicknesses indicate contribution ratios (weight x FC at each connection). Connections are labeled 'Positive/Negative (P/N)' based on correlation coefficient signs with letter 3-back learning performances, whereas numbers indicate descending orders of contribution ratio. Each network's color indicates relevance with working memory function based on BrainMap ICA (*Laird et al., 2011*); warmer colors indicate closer relevance to working memory function. See *Table 1* for the networks connected by the selected 16 connections, and precise values of contribution ratio of each connection. Each network's label and regions included in it are summarized in *Supplementary file 2*.

DOI: https://doi.org/10.7554/eLife.38844.003

The following figure supplement is available for figure 2:

**Figure supplement 1.** Letter 3-back learning curves for each participant.

*Figure 2 continued on next page*

*Figure 2 continued*

DOI: https://doi.org/10.7554/eLife.38844.004

## Prediction in independent test set of healthy individuals

We next tested the model's generalizability to an entirely independent healthy cohort using HCP dataset 500 Subjects Release (*Van Essen et al., 2013*). We restricted our analysis to participants for whom all rs-fMRI, visual-object N-back, the NIH Toolbox list-sorting test (*Tulsky et al., 2014*), and Raven's progressive matrices with 24 items (*Bilker et al., 2012*) were available (*N* = 474; 194 males, 5 year age ranges in the Open Access Data: 22–25, 26–30, 31–35 and 36 + years old). Individual working memory performance was briefly measured by the visual-object N-back with 0-back and 2-back conditions (visual-object N-back score) and the list-sorting test (*Figure 1B*). The N-back scores were evaluated by the accuracy percentage of 2-back and 0-back conditions (86.0 ± 9.5% (SD), range 45.8% to 100%). The other working memory measure, the list-sorting test, is a sequencing task of visual or auditory stimuli (mean scores: 110.5 ± 11.6 (SD), range 80.8 to 144.5). Additionally, general

**Table 1.** Selected connections and their contribution to working memory ability.

| Label | Connection (rank) | | Contribution ratio [%] |
|---|---|---|---|
| *Positive features* | | | |
| P1 | Left fronto-parietal network (1) | (within-network) | 33.9% |
| P2 | Supplemental motor network (3) | Primary sensorimotor network (hand) (11) | 15.4% |
| P3 | Middle frontal and parietal network (2) | Lateral temporal network (6) | 1.9% |
| *Negative features* | | | |
| N1 | Cingulo-opercular network (5) | Midbrain (10) | 13.9% |
| N2 | Right fronto-parietal network (4) | Midbrain (10) | 11.4% |
| N3 | Right fronto-parietal network (4) | Superior parietal network (18) | 9.2% |
| N4 | Supplemental motor network (3) | Orbitofrontal network (14) | 5.0% |
| N5 | Middle frontal and parietal network (2) | Primary sensorimotor network (mouth) (9) | 3.0% |
| N6 | Lateral occipital network (7) | Auditory (15) | 2.6% |
| N7 | Left fronto-parietal network (1) | Midbrain (10) | 1.8% |
| N8 | Cerebellum (12) | Auditory (15) | 1.3% |
| N9 | Left fronto-parietal network (1) | Lateral occipital network (7) | 0.3% |
| N10 | Lateral occipital network (7) | Primary sensorimotor network (hand) (11) | 0.3% |
| N11 | Lateral occipital network (7) | Superior parietal network (18) | 0.1% |
| N12 | Primary sensorimotor network (mouth) (9) | Cerebellum (12) | −0.0% |
| N13 | Left fronto-parietal network (1) | Basal ganglia (8) | −0.2% |

Rank indicates relevance with working memory function according to the BrainMap ICA.

DOI: https://doi.org/10.7554/eLife.38844.012

fluid intelligence was assessed by Raven's progressive matrices. The scores are integers that indicate the number of correct items (16.5 ± 4.8 (SD) from 4 to 24).

Before the generalization test, we found that the visual-object N-back task and the list-sorting test scores were positively correlated with general fluid intelligence (Spearman's rank correlation $\rho = 0.46$, p = 3.3 × 10$^{-26}$; $\rho = 0.32$, p = 5.7×10$^{-13}$, respectively) and negatively correlated with average in-scanner head motion (Spearman's rank correlation $\rho = -0.24$, p = 1.5 × 10$^{-7}$; $\rho = -0.12$, p = 0.009) as shown in *Figure 3—figure supplement 1*. To exclude these contaminations, we performed a partial correlation analysis while factoring out these two variables. This revealed a significant partial correlation of the predicted working memory ability with the measured visual-object N-back scores (Spearman's rank partial correlation $\rho = 0.11$; p = 0.0072, *Figure 3A*) and with the measured list-sorting scores (Spearman's rank partial correlation $\rho = 0.084$; p = 0.034). The model captures FC variations specific to working memory ability independently of general fluid intelligence and head motion.

Furthermore, we examined whether the model prediction was more similar to the 2-back score than the 0-back score. Spearman's rho partial correlation between the model prediction and task performance was 0.078 for 2-back task and 0.086 for 0-back task, while factoring out two confounding variables (fluid intelligence and head motion). There was no significant difference between the two correlation coefficients. Therefore, we could not conclude that the model prediction was more similar to 2-back score than 0-back score.

## Prediction in individual schizophrenia patients and controls

We examined whether the prediction model also predicted individual differences in working memory ability using independently collected resting state fMRI scans of schizophrenia (SCZ) dataset. The

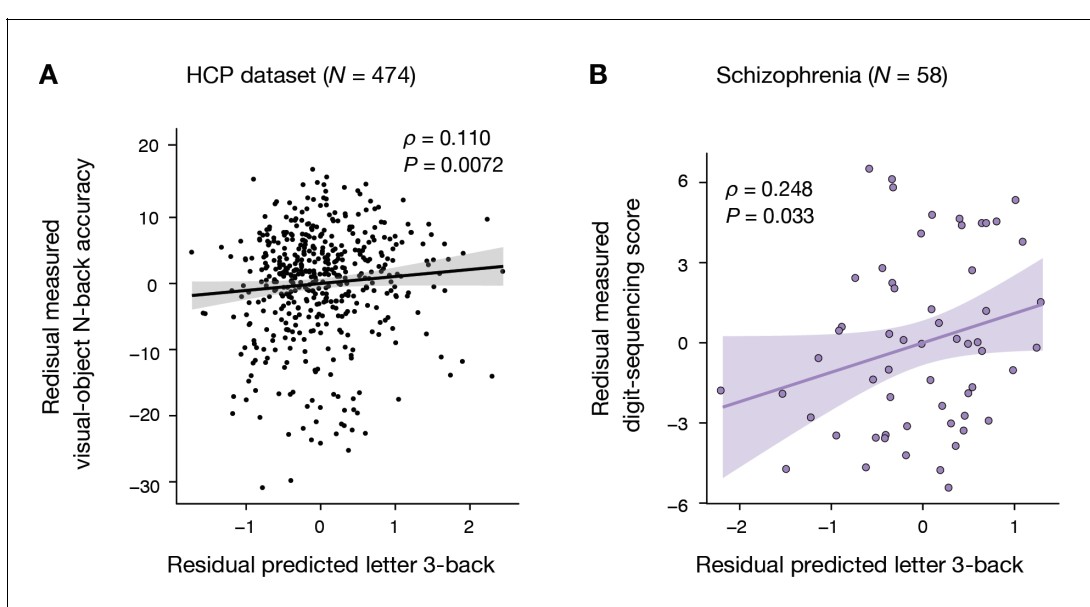

**Figure 3.** Generalizability to HCP dataset and schizophrenia dataset. (**A**) Significant Spearman's rank partial correlation between predicted letter 3-back learning performance and measured visual-object N-back accuracy while factoring out general fluid intelligence and head motion ($\rho = 0.110$, p = 0.0072). (**B**) Significant Pearson partial correlation between predicted letter 3-back performances and measured digit-sequencing scores while factoring out the composite BACS score and age ($\rho = 0.248$, p = 0.033).

DOI: https://doi.org/10.7554/eLife.38844.005

The following figure supplements are available for figure 3:

**Figure supplement 1.** Spearman's rank correlation matrix for HCP dataset.

DOI: https://doi.org/10.7554/eLife.38844.006

**Figure supplement 2.** Pearson's correlation matrices for schizophrenia samples.

DOI: https://doi.org/10.7554/eLife.38844.007

**Figure supplement 3.** Distribution of BACS digit-sequencing score.

DOI: https://doi.org/10.7554/eLife.38844.008

schizophrenia patients ($N$ = 58) and their age- and gender-matched controls ($N$ = 60) underwent a cognitive test battery the Japanese version of Brief Assessment of Cognition in Schizophrenia (BACS-J) (*Kaneda et al., 2007*). This test battery is composed of six subtests including a digit sequencing test as a working memory measure. In this test, auditory sequences of numbers were presented, with increasing length from three to nine digits (*Figure 1B*). Participants repeated the sequences aloud by sorting in ascending order. The digit-sequencing scores were the number of correct trials among 28 trials (18.4 ± 4.1 (SD), range 10 to 27 in patients while 22.9 ± 4.3 (SD), range 12 to 28 in controls). Their composite BACS score was evaluated by average score of BACS's five subtests other than the digit sequencing test.

First, we applied the model to patients with schizophrenia. Before the model application, we found that the digit-sequencing scores correlated positively with composite BACS score excluding working memory ($r$ = 0.61, p = $3.0 \times 10^{-7}$), negatively with age ($r$ = −0.36, p = 0.005), but not with head motion ($r$ = −0.03, p = 0.83) as shown in *Figure 3—figure supplement 2*. While controlling the age and the composite BACS score using a partial correlation analysis, the model predictions showed significant correlations with digit-sequencing scores ($\rho$ = 0.25, p = 0.033, *Figure 3B*). Second, we applied the model to full sample of SCZ patients ($N$ = 58) and controls ($N$ = 60). We found that the digit-sequencing score was correlated positively with composite BACS score excluding working memory ($r$ = 0.68, p = $2.0 \times 10^{-17}$), and negatively with age ($r$ = −0.36, p = $5.7 \times 10^{-5}$), but not with head motion ($r$ = −0.04, p = 0.68). While controlling the age and the composite BACS score, a partial correlation analysis showed that the model prediction is significantly correlated with the digit-sequencing score ($\rho$ = 0.15, p = 0.048). Therefore, the model captures FC variations that are specific to working memory ability independently of age or the composite BACS score.

Furthermore, we examined whether the model predictions were correlated with digit-sequencing score in controls alone. Before the model application, we found that the digit-sequencing scores distributed non-normally (Lilliefors test, p = 0.001) and correlated positively with composite BACS score excluding working memory (Spearman's rho = 0.52, p = $2.1 \times 10^{-5}$), but not with age (Spearman's rho = −0.19, p = 0.15) and head motion (Spearman's rho = 0.02, p = 0.88). While controlling the composite BACS score using a partial correlation analysis, the model predictions showed no significant correlations with digit-sequencing scores (Spearman's rho = −0.07, p = 0.60). This result is likely attributed to a ceiling effect in the BACS digit-sequencing score for controls (see *Figure 3—figure supplement 3*).

## Prediction in four distinct psychiatric disorders

We addressed whether our model could quantitatively reproduce degrees of working memory deficits across four psychiatric diagnoses, including schizophrenia (SCZ), major depressive disorder (MDD), obsessive-compulsive disorder (OCD), and autism spectrum disorder (ASD). Their demographic data are summarized in *Table 2*. This dataset were collected at a Japanese neuropsychiatry consortium (*Takagi et al., 2017*; *Yahata et al., 2016*) (https://bicr.atr.jp/decnefpro/). Previous studies generally observed working memory impairment, in descending order of severity, in SCZ, MDD, OCD and ASD (*Forbes et al., 2009*; *Lever et al., 2015*; *Snyder, 2014*; *Snyder et al., 2015*). We predicted individual working memory ability by applying the prediction model of working memory ability to their resting state functional connectivity. Then, we compared the model predictions between patients and age- and gender-matched controls scanned at the same site to remove differences in scanner and imaging protocols between sites. Consequently, we identified significant differences in the predicted working memory ability between the patients and controls only for SCZ patients (two-tailed $t$-test for SCZ group: $t_{116}$ = −3.68, P = ($3.5 \times 10^{-4}$) x 4 = 0.0014, Bonferroni corrected; *Figure 4A*). Next, we calculated individual patients' $Z$-score (normalized difference between a patient and average of controls at the same site) of the predicted working memory ability for each diagnosis (*Figure 4B*). A one-way ANOVA revealed a significant main effect of diagnosis on the $Z$-score ($F_{3,245}$ = 7.63, p = $6.8 \times 10^{-5}$). The severity of the predicted impairment in SCZ patients was larger than all other diagnoses (post-hoc Holm's controlled $t$-test, adjusted p < 0.05).

The predicted working memory ability alteration was more negative in the order of SCZ, MDD, OCD, and ASD with effect sizes (Hedge's $g$) of −0.68, –0.29, −0.16, and 0.09, respectively. Meta-analyses on working memory ability measured by digit span tasks (digit-span score) (*Forbes et al., 2009*; *Snyder, 2014*; *Snyder et al., 2015*) could provide a quantitative measure of working memory impairment for each diagnosis in terms of the effect size. Red horizontal lines in *Figure 4C* indicate

**Table 2.** Demographic data of the multiple psychiatric diagnoses dataset.

| Diagnosis | Site | Measure | Patients | Controls | Test | P-value |
|---|---|---|---|---|---|---|
| SCZ | KYU | N | 58 | 60 | - | - |
| | | Age | 37.9 | 35.2 | $t_{116} = 1.7$ | 0.1 |
| | | | (9.3) | (8.4) | | |
| | | Male % | 52% | 67% | Fisher's exact test | 0.13 |
| MDD | HRU | N | 77 | 63 | - | - |
| | | Age | 41.6 | 39.3 | $t_{138} = 1.3$ | 0.21 |
| | | | (11.2) | (12.0) | | |
| | | Male % | 56% | 46% | Fisher's exact test | 0.31 |
| OCD | KPM | N | 46 | 47 | - | - |
| | | Age | 32.2 | 30.3 | $t_{91} = 1.1$ | 0.28 |
| | | | (9.9) | (8.7) | | |
| | | Male % | 37% | 45% | Fisher's exact test | 0.53 |
| ASD | UTK (site1) | N | 33 | 33 | - | - |
| | | Age | 32.8 | 34.7 | $t_{64} = -1.0$ | 0.3 |
| | | | (8.4) | (7.0) | | |
| | | Male % | 64% | 55% | Fisher's exact test | 0.62 |
| | SHU (site2) | N | 36 | 38 | - | - |
| | | Age | 29.9 | 32.5 | $t_{72} = -1.5$ | 0.14 |
| | | | (7.2) | (7.4) | | |
| | | Male % | 100% | 100% | Fisher's exact test | 1 |
| | Pooled | N | 69 | 71 | - | - |
| | | Age | 31.3 | 33.5 | $t_{138} = -1.8$ | 0.08 |
| | | | (7.9) | (7.2) | | |
| | | Male % | 83% | 79% | Fisher's exact test | 0.67 |

**Site:** KYU, Kyoto University; HRU, Hiroshima University; KPM, Kyoto Prefectural University of Medicine; UTK, University of Tokyo; SHU, Showa University.
**Measure:** 'N' indicates the number of subjects; 'Age' is shown as mean (SD); 'Male %" is the fraction of male. The tests and p-values compare the patient and control groups within-site.
DOI: https://doi.org/10.7554/eLife.38844.011

confidence intervals of the effect sizes according to the meta-analysis studies. The effect sizes of the predicted working memory impairment fell within confidence intervals for forward digit-span in SCZ, MDD, and OCD and for backward digit-span in OCD (*Figure 4C*). Therefore, the model capturing a normal range of variation in working memory ability reproduced not only the order but also the quantitative aspects of working memory deterioration across the four distinct diagnoses. Note that no meta-analysis was available for ASD. However, previous studies generally showed little differences in verbal working memory ability from typically developed controls (*Koshino et al., 2005*; *Lever et al., 2015*; *Williams et al., 2005*), consistent with the predictions of our model. Moreover, we examined whether these predicted effect sizes were more similar to working memory ability or general cognitive ability (IQ) reported in meta-analysis studies (*Abramovitch et al., 2018*; *Ahern and Semkovska, 2017*; *Heinrichs and Zakzanis, 1998*). As illustrated in *Figure 4—figure supplement 1*, the predicted working memory ability falls within confidence interval of the IQ effect size only for first-episode MDD while it falls within confidence interval of effect size of working memory (forward digit span) for every diagnosis. Regarding the relative order of effect sizes, the effect size of IQ deficits can be ordered as SCZ, OCD, and MDD (first episode). In contrast, the effect sizes of working-memory deficits (as measured by digit-span task) can be ordered as SCZ, MDD and OCD, which is consistent with the order predicted by our model. Therefore, predicted working-memory deficits were more similar to observed deficits in working memory than those in fluid intelligence. We identified no significant differences in head motion between patients and their healthy

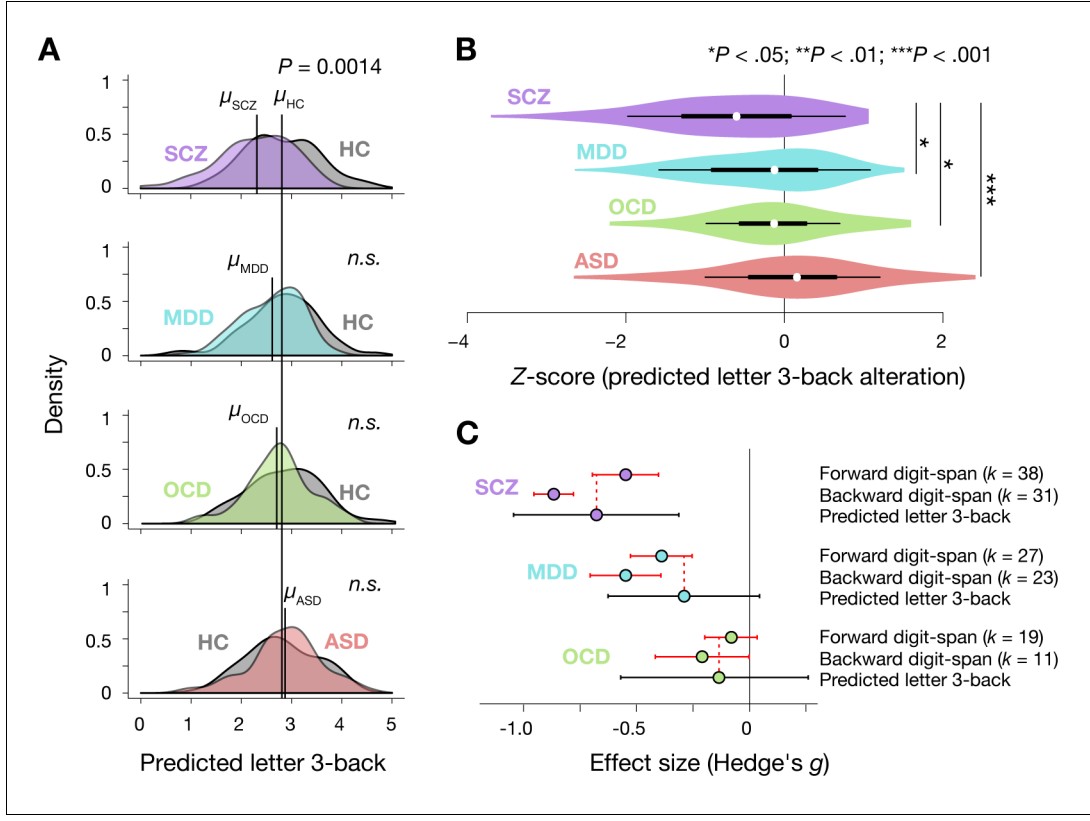

**Figure 4.** Prediction of diagnosis-specific alterations of working memory ability. (**A**) Predicted letter 3-back working memory ability for patients (N = 58, 77, 45, and 69 for SCZ, MDD, OCD, and ASD, respectively) and their age- and gender-matched healthy/typically developed controls (HC, N = 60, 62, 47, and 71) shown as kernel density. For illustration purposes, distribution of each control group was standardized to that of the ATR dataset, and the same linear transformation was applied to patients' distributions. $\mu$ indicates mean value for each group. (**B**) Violin plots of Z-scores for predicted working memory ability alterations. White circles indicate medians. Box limits indicate 25th and 75th percentiles. Whiskers extend 1.5 times interquartile range from 25th and 75th percentiles. (**C**) Comparison of estimated effect sizes for working memory deficits. k indicates number of studies included in the meta-analyses (*Forbes et al., 2009*; *Snyder, 2014*; *Snyder et al., 2015*). Error bars indicate 95% confidence intervals.

DOI: https://doi.org/10.7554/eLife.38844.009

The following figure supplement is available for figure 4:

**Figure supplement 1.** Effect sizes of IQ, digit-span, and predicted working memory ability.
DOI: https://doi.org/10.7554/eLife.38844.010

controls in any diagnosis (two-tailed t-tests, at largest $t_{138} = 1.77$, p > 0.080 observed in ASD group).

## Functional connectivity patterns in psychiatric diagnoses

Given a common FC-WMA relationship across these diagnoses, we examined how diagnosis-dependent working memory impairment resulted from FC alteration patterns. Since the model is the weighted summation of 16 FC values, increased/decreased working memory ability is determined by the sum of the increased/decreased weighted-FC values of the connections. To investigate the effect of FC alteration at each connection on working memory impairment, for each individual patient, we examined the difference in weighted-FC value at each connection from the average of the corresponding controls. We call this difference the D-score (see Materials and methods). By averaging the D-scores within each diagnosis, *Figure 5A* shows how the accumulation of diagnosis-dependent D-scores resulted in difference in working memory impairment across the diagnoses. Some D-scores are relatively constant, while others are variable across diagnoses. For example, the D-score for P1

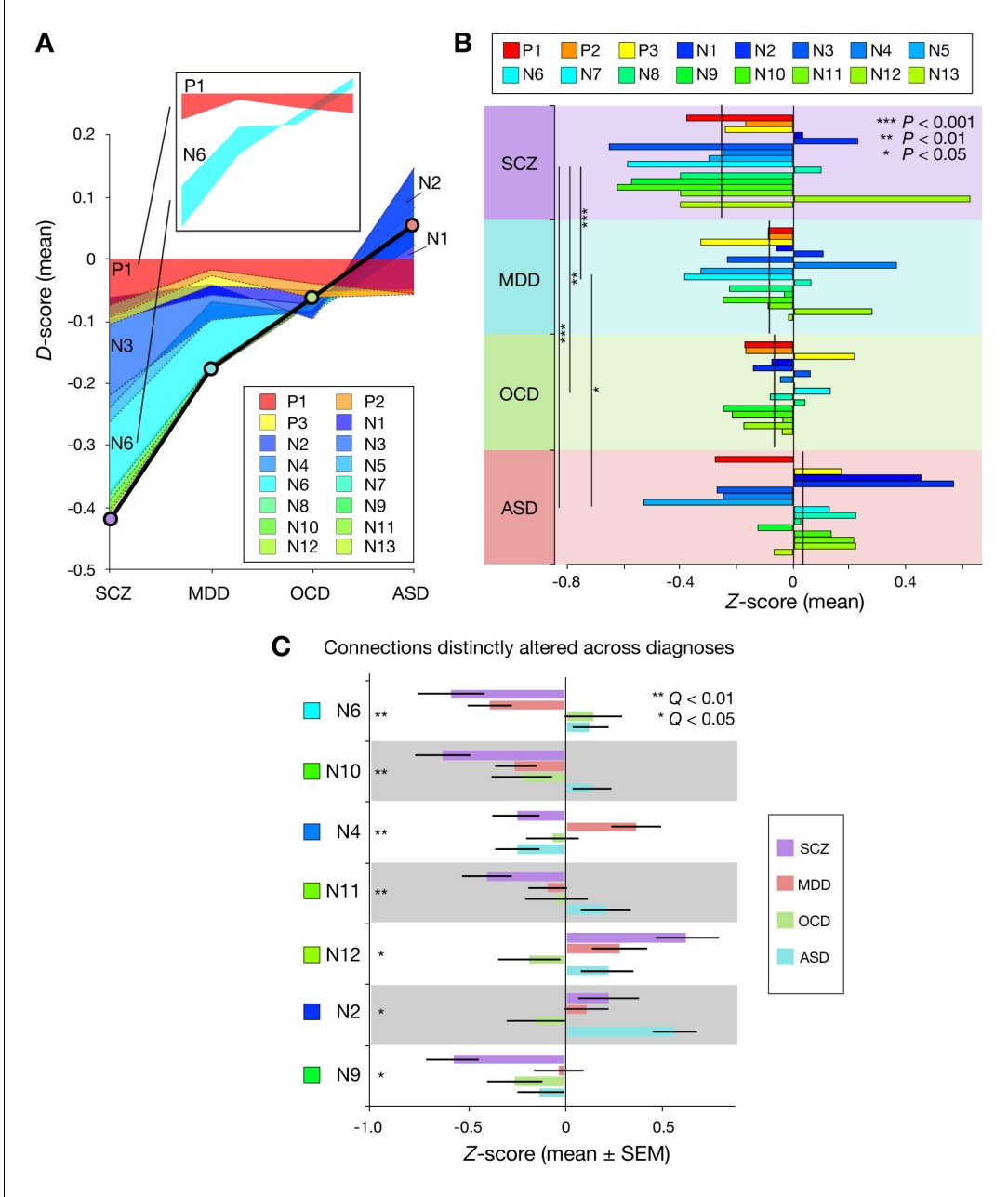

**Figure 5.** Accumulation of function connectivity differences exhibits diagnosis-specific working memory ability. (**A**) Accumulation of averaged *D*-scores for all 16 connections. Bold black line indicates summation of contributions by all connections, corresponding to predicted working memory ability alteration. This figure shows how diagnosis-specific working memory impairment results from complex disturbances of multiple connections. Upper panel depicts two representative alteration patterns across diagnoses. While connection P1 commonly decreased working memory ability across diagnoses, connection N6 distinctly affected working memory ability (decrease in SCZ and MDD and increase in OCD and ASD). (**B**) *Z*-scores (normalized *D*-scores) for each diagnosis. Left asterisks and lines indicate significant differences in mean *Z*-scores between two diagnoses (p < 0.05, Bonferroni corrected). Vertical lines across horizontal bars indicate *Z*-scores averaged across connections. (**C**) *Z*-scores for connection that showed a significant effect of diagnosis. Connections were sorted by small p values of diagnosis effect (Kruskal-Wallis test, *Q* < 0.05, FDR corrected).
DOI: https://doi.org/10.7554/eLife.38844.013

The following figure supplements are available for figure 5:

**Figure supplement 1.** Connections of non-significant effect of diagnosis.
DOI: https://doi.org/10.7554/eLife.38844.014

**Figure supplement 2.** Distribution of magnitude of diagnosis effect on connectivity change between the connections in the model and other connections.

*Figure 5 continued on next page*

*Figure 5 continued*

DOI: https://doi.org/10.7554/eLife.38844.015

**Figure supplement 3.** *D*-scores related to network clusters.

DOI: https://doi.org/10.7554/eLife.38844.016

was commonly negative regardless of the diagnosis, while the *D*-score for N6 was negative or positive, dependent on the diagnosis (inset, *Figure 5A*). Therefore, *Figure 5A* qualitatively suggests that diagnosis-dependent working memory impairment is derived from complex FC alterations patterns.

To compare the weighted FC-values between the diagnoses, we calculated standardized score (*Z*-score) by dividing *D*-score by the standard deviation of the corresponding control group (see Materials and methods). We entered the *Z*-score in a two-way ANOVA with diagnosis as a between-participant factor and connection as a within-participant factor. We found a significant main effect of diagnosis ($p < 1.0 \times 10^{-4}$; *Figure 5B*). SCZ patients showed significantly more negative mean *Z*-scores across connections than the other diagnoses (post-hoc diagnosis-pair-wise comparisons, $p < 0.05$). This suggests that the global patterns of the working memory ability-related 16 connections in SCZ were more severely disrupted than the other three diagnoses.

We found a significant interaction effect between diagnosis and connection in the *Z*-scores ($p < 1.0 \times 10^{-5}$), suggesting that FC alterations at particular connections are diagnosis-dependent. In seven connections (*Figure 5C*), the *Z*-scores were significantly different among the four diagnoses ($Q < 0.05$, false discovery rate (FDR) corrected), suggesting that these connections are differentially altered across the diagnoses. Conversely, no significant differences in the *Z*-scores were observed across diagnoses in the remaining nine connections (*Figure 5—figure supplement 1*).

Furthermore, we examined whether the working memory ability-related 16 connections were more consistently altered in patients relative to controls than connections excluded from the model. Specifically, we first quantified the effect of diagnostic labels on connectivity alterations for every connection by using chi-square values of a Kruskal-Wallis test (a non-parametric version of one-way ANOVA). Then, we tested if the distribution of the chi-square values was different between the model's 16 connections and other 155 connections (Kolmogorov–Smirnov test). Consequently, we found no significant difference between the distributions ($p = 0.30$). This means that connectivity is altered at the working memory ability-related connections as well as the other connections (*Figure 5—figure supplement 2*).

To understand these results from global brain networks, we grouped the 18 networks into seven clusters based on the hierarchical clustering of the networks performed in the BrainMap ICA study (*Laird et al., 2011*). They were named fronto-parietal, motor/visuospatial, emotion/interoception, audition/speech, visual, cerebellum and default-mode clusters (*Supplementary file 2*). We selected connections bridging between different clusters. We then fixed a cluster and summed *D*-scores (averaged across participants of each diagnosis) of the connections that have nodes (networks) in the cluster. This summation was repeated for every cluster. *Figure 5—figure supplement 3* shows the summed *D*-score for each cluster and diagnosis. The four diagnoses commonly showed altered connectivity related to the fronto-parietal cluster. This confirmed importance of the fronto-parietal networks across diagnosis. The motor/visuospatial, audition/speech, and visual clusters are associated with lower working memory in schizophrenia and MDD. This suggests that dysfunctions in motor and sensory systems are related to lower working memory in specific diagnoses. Note that we could not find any connections that have a node in the default-mode cluster, which does not appear in the figure.

## Discussion

We built a prediction model of working memory ability using data-driven analysis of whole-brain connectivity among healthy Japanese individuals. Our model predicted individual differences of working memory ability in SCZ patients. It also reproduced the order of working memory impairment for four distinct diagnoses (i.e., SCZ > MDD > OCD > ASD). Moreover, the magnitudes of reproduced impairment were consistent with previous meta-analyses. Our results provide the first evidence for a common whole-brain FC-WMA relationship across healthy populations and a range of psychiatric disorders. That is, our results support the idea that working memory impairment in

psychiatric disorders is a continuous deviation from a normal pattern while preserving the common relationship between brain-wide connectivity and working memory ability. Our detailed examination suggested that the difference in degrees of the impairment across the diagnoses results from both common and diagnosis-specific connectivity changes within the common FC-WMA relationship.

Our model's generalizability to completely independent datasets is supported by rigorous methods. Our simple model by combining low-number (=18) of network nodes and sparse estimation of relevant FC compensated the weakness of relatively small training sample size (=17). Also, our careful evaluation of working memory ability in 1500 trials devoting about 80–90 min enhanced measurement precision by reducing trial-by-trial variability at the individual participant level (*Smith and Little, 2018*). In this way, our training dataset ensured that our model captured low-complexity FC pattern essential for individual working memory ability. We note that larger sample sizes do not really always improve prediction accuracy. Using identical methods in our model construction, we developed a new prediction model of visual-object N-back score using the HCP dataset as a training dataset ($N = 474$). However, this model failed to provide significant prediction even within the training samples ($R^2 = 0.005$). This seemingly unexpected result may partly result from differences in the way where individuals' working memory ability were evaluated. The HCP conducted the N-back task in a limited time (160 trials for each individual), which may be noisy to precisely characterize individual ability. On the other hand, Rosenberg et al. built a model with careful examination of cognitive performance (more than 40 min of attention task), using modest ($N = 25$) training samples and demonstrated robust generalization to independent test sets (*Rosenberg et al., 2016*). Our model's accuracy was comparable with their model of attention ability for an external test set ($r \sim 0.3$).

We carefully excluded the spurious correlations (*Siegel et al., 2016*; *Whelan and Garavan, 2014*). We examined general intellectual/cognitive ability, age, and head motion and confirmed that these disturbance variables had a minimal effect on prediction. Moreover, we analyzed age- and gender-matched controls from the same sites and compared the alterations from the controls (*Z*-scores), thereby minimizing the false positives that could be derived from age, gender, or imaging sites/parameters.

Our proposed two-stage approach, which builds a normative model and applies it to multiple diagnoses, is an effective technique to systematically compare neural substrates across multiple diagnoses. Clinical measures of attention deficit hyperactivity disorder were previously predicted by FC patterns that determine attention ability in healthy populations (*Rosenberg et al., 2016*), suggesting common connectivity-cognition relationships across healthy and clinical populations. By extending this approach, we directly examined working memory ability of the patients in cognitive tasks rather than assessments of clinical symptoms based on subjective report or behavioral observation. We also tested the model across not only a single diagnosis but also multiple diagnoses.

By coherently establishing FC-cognition relationships from normal to abnormal, our two-stage approach could potentially cluster multiple psychiatric disorders based on neurobiological measures and behaviors (*Insel et al., 2010*). Such neurobiological insights into behavioral abnormality are consistent with recent transdiagnostic studies of genomics (*Cross-Disorder Group of the Psychiatric Genomics Consortium, 2013*; *O'Donovan and Owen, 2016*; *Plomin et al., 2009*) and neuroimaging (*Clementz et al., 2016*; *Goodkind et al., 2015*; *Sheffield et al., 2017*), which indicate that some neurobiological changes are shared across psychiatric diagnoses. Consistent with our results, recent studies provide evidences indicating that models of attention and memory generalize to predict behavior in patient as well as in healthy populations (*Kessler et al., 2016*; *Lin et al., 2018*; *O'Halloran et al., 2018*; *Rosenberg et al., 2016*).

Our results identified alterations in large-scale network clusters that correlated with working memory impairment (*Figure 5—figure supplement 3*). First, the four diagnoses commonly showed altered connections related to the fronto-parietal networks. This finding support a hypothesis that the executive control network regulates symptoms, and its dysregulation is a shared neural substrate across diagnostic categories (*Cole et al., 2014*; *Smucny et al., 2018*). Second, visual and auditory networks were associated with lower working memory ability in schizophrenia and MDD. These two results are consistent with the hypothesis that cognitive function is disrupted regarding not only top-down executive control but also bottom-up sensory processes (*Javitt, 2009*). Recently, a neurophysiological study has suggested contributions of motor and premotor neurons to encoding serial order of working memory (*Carpenter et al., 2018*). This is consistent with our result that alterations of

connections related to the motor/visuospatial networks were associated with lower working memory ability in schizophrenia, MDD, and ASD.

The test-retest reliability of our functional connectivity estimation methods was fair level (0.2 < ICC ≤ 0.4) according to an interpretation criteria of ICC (*Landis and Koch, 1977*). A previous study on test-retest reliability of functional connectivity between 18 different brain regions (*Birn et al., 2013*), reported similar ICC values (ICC ~0.2) when scan length was 6 to 15 min. Another previous study examined functional connectivity reliability (*Noble et al., 2017*), using 268 regions from whole-brain, also reported that 6 min of scan length yielded similar reliability (dependability coefficient ~0.2 to 0.4). Although our connectivity estimation methods cannot reach clinically recognized request (ICC > 0.8), these studies suggest that test-retest reliability of our methods are comparable to other common connectivity estimation methods.

Regarding the large contribution of the left fronto-parietal network (FPN) to our prediction model in comparison to the right FPN, the BrainMap ICA on which our network definition is based gives us useful information. The BrainMap ICA paper (*Laird et al., 2011*) reported that IC18 (left FPN) has greater functional relevance to working memory than IC15 (right FPN) based on meta-analyses of thousands of publications. Moreover, a meta-analysis on N-back task with different stimulus modality (*Owen et al., 2005*) found monitoring of verbal stimuli was strongly associated with left ventrolateral prefrontal cortex (a part of left FPN), while monitoring of spatial locations activated right lateralized frontal and parietal regions. In the current study, we used a letter 3-back task that requires encoding alphabet letters, which are more related to word monitoring than location monitoring. Therefore, the left FPN would be expected to contribute more to our prediction model than the right FPN. Although large portion of the model relies on the within-network connectivity of the left FPN (~34% contribution), the right FPN also showed a substantial contribution to working memory via negative connectivity N2 (connection with the midbrain network, please see *Figure 2*) and N3 (connection with the superior parietal network) (~20% contribution).

The primary limitation of this study is the assumption that our model captures general capability of working memory not restricted to letter 3-back performance. Working memory is an umbrella term which involves multiple distinct sensory modalities and executive functions, and the empirical findings and theoretical conceptualization is still rapidly extending (*Chatham and Badre, 2015*; *Cogan et al., 2017*; *D'Esposito and Postle, 2015*; *Ma et al., 2014*; *Myers et al., 2017*; *Serences, 2016*). Rather than focus a single specific domain, we utilized any domains of working memory performance (letter N-back, visual object N-back, digit-sequencing task, and digit span). Future work may reveal more elaborate findings for FC-WMA relationships based on more nuanced definition of WMA, since distinct types of working memory tasks are engaged with specific neural processes (*Nee et al., 2013*; *Owen et al., 2005*). Second, working memory performance was measured only in schizophrenia patients but not in other diagnostic groups. Therefore, it was impossible to compare their predicted working memory ability with measured scores. This presents challenges for the between-group comparisons in the patient samples. Third, although the participants are matched on age and sex within each site, the groups may differ along a number of dimensions beyond working memory (e.g. medication status, scanning protocol, and potentially IQ and other cognitive abilities). It is difficult to fully control every dimension, and little is known how such dimensions affect estimation of functional connectivity. Fourth, the results in the HCP dataset showed that only a little variance can be explained by our model. This may be attributed to considerable differences between the HCP dataset and the ATR dataset. The major differences include population location (the American vs. the Japanese), and working memory task properties that contrast in sensory modality (visual object vs. verbal), number of observations made for each individual (160 vs. 1500 trials), difficulty level (0-back and 2-back vs. 3-back), and measurement environment (in vs. out of MRI scanner). Finally, in the HCP dataset, the model predictions were not more closely related to 2-back than 0-back performance. This result suggest that the model may capture abilities beyond working memory.

In conclusion, our data provide a unified working memory ability framework across healthy populations and multiple psychiatric disorders. Our whole-brain functional connectivity model quantitatively predicted individual working memory ability in independently collected cohorts of healthy populations and patients with any of four psychiatric diagnoses (N = 965). Our results suggest that the FC-WMA relationship identified in healthy populations is commonly preserved in these psychiatric diagnoses and that working memory impairment in a range of psychiatric disorders can be

explained by the cumulative effect of multiple disturbances in connectivity among distributed brain networks. Our findings lay the groundwork for future research to develop a quantitative, brain-wide-connectivity-based prediction model of human cognition that spans health and psychiatric disease.

# Materials and methods

## ATR dataset

This dataset was a final set of participants after excluding individuals who exhibited noisy data collected in our previous study (*Yamashita et al., 2015*). Here, we used this dataset to construct a normative prediction model of working memory ability regarding a letter 3-back task learning plateau (N = 17, age 19–24 years old, 11 males). Recent study on predictive modeling of a single specific task performance using fMRI connectivity has reported comparable sample size (*Baldassarre et al., 2012*; *Rosenberg et al., 2016*).

### Working memory assessment

The participants performed a letter 3-back task (*Figure 1A*) over 25 sessions of training, with 60 trials for each session (1500 trials in total training sessions taking about 80–90 min). We obtained an individual learning curve by calculating the d-prime for each session, and by smoothing the data points with five-session moving average (*Figure 2—figure supplement 1*). The individual learning curve was fitted by an inverse curve ($y = a - b/x$), where $y$ is a d-prime in the $x$-th session, while $a$ and $b$ is a parameter for learning plateau and learning speed, respectively. We used the estimated learning plateau ($a$) for a measure of individual working memory ability (letter 3-back WMA). More detailed information is described in our previous paper (*Yamashita et al., 2015*).

### Functional connectivity estimation

We recorded a rs-fMRI scan with $3 \times 3 \times 3.5$ mm spatial resolution and a temporal resolution of 2.0 s for each participant (5 min 4 s). After removing the first two volumes, the data were preprocessed with slice timing correction, motion correction, and spatial smoothing with an isotropic Gaussian kernel (full width at half maximum = 8 mm). To remove several sources of spurious variance, we regressed out six motion parameters and the averaged signals over gray matter, white matter, and cerebrospinal fluid (*Fox et al., 2005*). The gray matter signal regression improves FC estimation by effectively removing motion-related artifacts (*Burgess et al., 2016*; *Ciric et al., 2017*; *Power et al., 2014*). Finally, we performed 'scrubbing' (*Power et al., 2012*) in which we removed scans where framewise displacement was > 0.5 mm.

We used network-level rather than node-level connectivity features to avoid overfitting to training samples (the curse of dimensionality). Specifically, we calculated FC values, based on the 18 whole-brain intrinsic networks of BrainMap ICA (*Laird et al., 2011*), for pair-wise between-network ($18 \times 17/2 = 153$) connections and within-network connections. Between-network FC was calculated as Pearson's correlation between blood-oxygen-level dependent signal time courses averaged across voxels within each network. Within-network FC was calculated as mean voxel-wise correlations within each of 18 networks.

### Test-retest reliability of functional connectivity estimation

To examine test-retest reliability of the functional connectivity estimation method, we calculated intra-class correlation (ICC) using three different datasets from Consortium for Reliability and Reproducibility (*Zuo et al., 2014*). We picked up following three datasets, Beijing Normal University (BNU 1), Institute of Automation, Chinese Academy of Sciences (IACAS 1), and University of Utah (Utah 1). We selected these datasets because 1) they have test-retest data across fMRI sessions, 2) ages of participants are comparable with those in our discovery dataset that was used for the construction of our model (ATR dataset), 3) two datasets include Asian participants (participants in ATR dataset are Japanese). BNU one includes data from 57 healthy young volunteers (age 19–30 years, 30 males) who completed two MRI scan sessions within an interval of approximate 6 weeks (33–50 days, mean 40.94 days). All were right-handed and had no history of neurological and psychiatric disorders. The resting state fMRI data was collected for 6 min 46 s. Detailed information is available for BNU one at http://fcon_1000.projects.nitrc.org/indi/CoRR/html/bnu_1.html. Seven participants ('BNU25914' to

'BNU25920') have incomplete data, thus these data were not used for the analysis. IACAS one includes data from 28 healthy young volunteers (age 19–43 years, 13 males) who completed two MRI scan sessions within an interval of approximate 6 weeks (20–343 days, mean 75.2 days). The resting state fMRI data was collected for 8 min. Detailed information is available for IACAS one at http://fcon_1000.projects.nitrc.org/indi/CoRR/html/iacas_1.html. Utah 1 includes 26 healthy young volunteers (age 8–39 years; 26 males) who completed two MRI scan sessions at least two years apart (733–1,187 days, mean 928.4 days). The resting state fMRI data was collected for 8 min 4 s. Detailed information is available for Utah one at http://fcon_1000.projects.nitrc.org/indi/CoRR/html/utah_1. html. To estimate functional connectivity, we used the same analysis pipeline. We obtained 171 (153 between-network and 18 within-network) functional connectivity values for each participant/session (test or retest). To estimate test-retest reliability, intra-class correlation (ICC) was calculated for each of the 171 functional connectivity values (univariate test-retest reliability). ICC was calculated by following equation:

$$\mathrm{ICC} = (\mathrm{MS_b} - \mathrm{MS_w})/\{\mathrm{MS_b} + (k-1)\,\mathrm{MS_w}\}$$

where, $\mathrm{MS_b}$ is the between-subjects mean squared error and $\mathrm{MS_w}$ is the within-subjects mean squared error and $k$ is the number of independent fMRI measures (i.e. $k = 2$ for test and retest). We put negative ICC values to be zeros as done by previous studies (e.g. *Zhang et al., 2011*).

## Developing prediction model

To predict individual learning plateaus in the letter 3-back task, we performed a sparse linear regression analysis (*Sato, 2001*) on the whole-brain FC values (http://www.cns.atr.jp/cbi/sparse_estimation/sato/VBSR.html). Individual working memory ability was modeled as a linear weighted summation of FC values at a small number of connections among the intrinsic networks. The connections were automatically selected by the sparse linear regression algorithm. In our previous study (*Yamashita et al., 2015*), we employed a leave-one-out cross-validation to estimate the prediction accuracy, and the analysis achieved high prediction accuracy within this dataset ($R^2 = 0.73$). To build a single prediction model, we utilized all the data ($N = 17$) as the training set.

## Human connectome project (HCP) dataset

The dataset was collected in the HCP and shared as 500 Subjects Release (*Van Essen et al., 2013*). We restricted our analysis to participants for whom all rs-fMRI, visual-object N-back, the NIH Toolbox list-sorting test (*Tulsky et al., 2014*), and Raven's progressive matrices with 24 items (*Bilker et al., 2012*) were available ($N = 474$; 194 males, 5 year age ranges in the Open Access Data: 22–25, 26–30, 31–35 and 36 + years old).

## Working memory assessment

Individual working memory performance was briefly measured by the visual-object N-back with 0-back and 2-back conditions (visual-object N-back score) and the list-sorting test (*Figure 1B*). The N-back task was performed in two fMRI runs, and each run contains eight task blocks of 10 trials (80 trials for each 0-back and 2-back condition). The scores were evaluated by the accuracy percentage of 2-back and 0-back conditions (86.0 ± 9.5% (SD), range 45.8% to 100%). The other working memory measure, the list-sorting test, is a sequencing task of visual or auditory stimuli (mean scores: 110.5 ± 11.6 (SD), range 80.8 to 144.5). Additionally, general fluid intelligence was assessed by Raven's progressive matrices. The scores are integers that indicate the number of correct items (16.5 ± 4.8 (SD) from 4 to 24).

## Examination of model prediction

We used rs-fMRI data (2 mm isotropic spatial resolution and a temporal resolution of 0.72 s) that were pre-processed and denoised by a machine learning tool that removes structured noise and moment-to-moment motion parameters (*Salimi-Khorshidi et al., 2014*). Additionally, spatial smoothing (full width at half maximum = 4 mm) and nuisance regression was performed using average signals of gray matter, white matter, and cerebro-spinal fluid. To extract slow oscillation and removes high-frequency noise (e.g. cardiac pulsation around 0.3 Hz), a band-pass filter (0.009–0.08 Hz) was applied and volumes with framewise displacement > 0.5 mm were removed. After

preprocessing of fMRI data, FC was estimated following procedures described in the ATR dataset. We entered FC values into the prediction model developed from the ATR dataset, and predicted individual working memory ability. Because the scores of the visual-object N-back task and list-sorting test showed non-normal distributions, we used a nonparametric statistical test (Spearman's rank correlation) to examine the model prediction accuracy. We compared the rank correlation coefficient between the predicted working memory ability and actual working memory scores with the null distribution obtained by shuffling participant labels (10,000 permutations). Links between behavioral scores and motion measures were preserved. Specifically, the subject label was shuffled for predicted working memory ability while the subject labels were preserved for confounding factors (e.g. age, fluid intelligence, and head motion).

## Multiple psychiatric diagnoses dataset

These data were collected at a Japanese neuropsychiatry consortium (*Takagi et al., 2017*; *Yahata et al., 2016*; *Yahata et al., 2017*; *Yamada et al., 2017*; *Yamashita et al., 2018*) (https://bicr-resource.atr.jp/decnefpro/). Resting state fMRI analysis was performed in the same way as described in ATR dataset. We performed slice timing correction and then motion estimation. The estimated motion parameters were used to estimate excessive motion data by frame-wise displacement > 0.5 mm. We did not remove a frame before or after the excessive motion. We conducted quality control for the rs-fMRI data and excluded participants if more than 40% of their total number of volumes of their data were removed by the scrubbing method. We calculated the ratio of excluded volumes to the total number of volumes for each subject, and averaged within patients or controls for each diagnosis. They were $2.3 \pm 5.5$ % / $1.4 \pm 2.9$% (patients/controls) for SCZ, $2.7 \pm 6.6$ % / $2.4 \pm 6.3$% for MDD, $0.4 \pm 0.8$ % / $0.7 \pm 1.7$% for OCD, and $1.4 \pm 3.8$ % / $4.6 \pm 8.5$% for ASD. We found a significant difference in the ratio between patients and controls only for ASD ($t_{97.3} = 2.91$, p = $4.4 \times 10^{-3}$). We detected outliers within each group (defined as values > 3 SD from the mean) for a control participant of MDD and a patient with OCD ($N = 1, 1$, respectively). These two participants were excluded from further analysis. After their data quality was assured, age- and gender-matched healthy control subjects were included in the analysis. Consequently, we used the rs-fMRI data of patients with SCZ, MDD, OCD, and ASD ($N = 58, 77, 46$, and 69, respectively) as well as their age- and gender-matched healthy/typically developed controls ($N = 60, 63, 47$, and 71). These sample sizes were comparable to or even larger than recent generalization test analysis (*Rosenberg et al., 2016*). Demographic data is summarized in *Table 2*. Scanning parameters are reported in *Supplementary file 1*.

### Participants with SCZ

Patients with SCZ diagnosed with the patient edition of the Structured Clinical Interview for the Diagnostic and Statistical Manual of Mental Disorders, Fourth Edition (DSM-IV) Axis I Disorders (SCID) (*First et al., 1995*) were recruited from in- and out-patients facilities in the Kansai region, Japan. The controls had no history of psychiatric illness, as screened with the non-patient edition of the SCID (*First et al., 2002*), and it was confirmed that their first-degree relatives had no history of psychotic disorders. Exclusion criteria for all individuals included a history of head trauma, neurological illness, serious medical or surgical illness, and substance abuse. All participants were physically healthy when they undertook the scanning. All the patients with SCZ had received antipsychotic medication. The mean ± SD values of medications based on chlorpromazine equivalents were $608 \pm 459$ mg/day. Other medications that the patients received were as follows; antiparkinsonism drugs ($N = 23$), anxiolytics and sleep inducing drugs ($N = 39$). The study design was approved by the Committee on Medical Ethics of Kyoto University and was conducted in accordance with the Code of Ethics of the World Medical Association. After being given a complete description of the study, all participants gave written informed consent.

### Participants with MDD

Patients with MDD were recruited from local clinics in Hiroshima, Japan, and all the patients were screened with the DSM-IV criteria for a unipolar MDD diagnosis, using the mini-international neuropsychiatric interview (M.I.N.I.) (*Otsubo et al., 2005*; *Sheehan et al., 1998*). No patient had current or past SCZ episodes. Healthy participants were recruited from the local community. They were

interviewed with the M.I.N.I. and none showed a history of psychiatric disorders according to DSM-IV criteria. At the time of scanning, six MDD individuals were medication free, and the rest of MDD individuals had been administered the following psychotropic drugs: antidepressants ($N = 70$), antipsychotics ($N = 7$), antiepileptics ($N = 7$), anxiolytics ($N = 15$), and sleep inducing drugs ($N = 26$), before the scanning. Around half of participants had been administered multiple drugs ($N = 36$). The current study protocol was approved by the Ethics Committee of Hiroshima University. Prior to the administration of any experimental procedure, written informed consent was obtained from all the participants.

## Participants with OCD

Patients were recruited at the Kyoto Prefectural University of Medicine Hospital, Kyoto, Japan. All patients were primarily diagnosed as OCD using the Structured Clinical Interview for DSM-IV Axis I Disorders-Patient Edition (SCID) (*First et al., 1995*). Exclusion criteria were 1) cardiac pacemaker or other metallic implants or artifacts; 2) significant disease, including neurological diseases, disorders of the pulmonary, cardiac, renal, hepatic, or endocrine systems, or metabolic disorders; 3) prior psychosurgery; 4) DSM-IV diagnosis of mental retardation and pervasive developmental disorders based on a clinical interview and psychosocial history; and 4) pregnancy. We excluded patients with current DSM-IV Axis I diagnosis of any significant psychiatric illness except OCD as much as possible and only four patients with trichotillomania, one patient with tic disorder, and one patient with tic disorder and specific phobia were included as patients with comorbidity. Thirty-five OCD individuals and 34 healthy controls were also included in a published paper (*Abe et al., 2015*).

There was no history of psychiatric illness in the healthy controls as determined by the Structured Clinical Interview for DSM-IV Axis I Disorders, Non-patient Edition (SCID-NP) (*First et al., 2002*). Additionally, we confirmed that there was no psychiatric treatment history in any of their first-degree relatives. At the time of scanning, 40 OCD individuals were medication free, whereas the remaining five OCD individuals had been administered the following psychotropic drugs: anxiolytics ($N = 2$), antidepressants ($N = 5$), before the scanning. Some participants had been administered multiple drugs ($N = 2$). The Medical Committee on Human Studies at the Kyoto Prefectural University of Medicine approved all the procedures in this study. All participants gave written, informed consent after receiving a complete description of the study.

## Participants with ASD (site 1)

Patients with ASD were recruited through the Department of Child Psychiatry and Neuropsychiatry at the University of Tokyo Hospital and via an advertisement on the website of the University of Tokyo Hospital. All ASD participants ($N = 33$) were diagnosed with pervasive developmental disorder based on the DSM-IV-TR criteria (*Association, 2000*). DSM-IV-TR diagnoses of autistic disorder, Asperger's disorder, or pervasive developmental disorder not otherwise specified ($N = 22$, $N = 3$, and $N = 8$, respectively) were supported by Autism Diagnostic Observation Schedule (*Lord et al., 1994*) ($N = 33$) and Autism Diagnostic Interview-Revised (Catherine Lord, Rutter, & Le Couteur, 1994) ($N = 25$). The Japanese version of M.I.N.I. (*Otsubo et al., 2005*; *Sheehan et al., 1998*) was used to evaluate psychiatric comorbidity. No participant satisfied the diagnostic criteria for substance use disorder, bipolar disorder, or SCZ. The intelligence quotient (IQ) scores of participants with ASD were obtained using the Wechsler adult intelligence scale-revised (WAIS-R) or third edition (WAIS-III). The full-scale IQs of all of the individuals with ASD were measured and found to be greater than 85. Typically-developed individuals were recruited from the local community. M.I.N.I. was used to confirm that none of the typically developed individuals met the diagnostic criteria for any psychiatric disorder. The IQs of the typically developed individuals were estimated using the Japanese version of the national adult reading test (*Matsuoka et al., 2006*). All participants were right-handers according to the Edinburgh Handedness Inventory (*Oldfield, 1971*). They completed the Japanese version of the autism-spectrum quotient (*Wakabayashi et al., 2007*). At the time of scanning, 10 ASD individuals were medication free, whereas the remaining 23 ASD individuals had been administered the following psychotropic drugs: anxiolytics ($N = 17$), antidepressants ($N = 19$), antipsychotics ($N = 15$), antiepileptics ($N = 5$), and sleep inducing drugs ($N = 17$), before the scanning. Some participants had been administered multiple drugs ($N = 19$). All participants provided

written informed consent as approved by The Ethics Committee of the Graduate School of Medicine and Faculty of Medicine at the University of Tokyo.

### Participants with ASD (site 2)

Patients with ASD were recruited from outpatient units of the Karasuyama Hospital, Tokyo, Japan. A team of three experienced psychiatrists and a clinical psychologist assessed all patients. All patients were diagnosed with ASD based on the criteria of the DSM-IV (*Association, 2000*) and a medical chart review. The assessment consisted of participant interviews about developmental history, present illness, life history, and family history and was performed independently by a psychiatrist and a clinical psychologist in the team. Patients were also asked to bring suitable informants who had known them in early childhood. At the end of the interview, the patients were formally diagnosed with a pervasive developmental disorder by the psychiatrist if there was a consensus between the psychiatrist and clinical psychologist; this process required approximately three hours. The group of typically developed individuals was recruited by advertisements and acquaintances. None of the typically developed individuals reported any severe medical problem or any neurological or psychiatric history. None of them satisfied the diagnostic criteria for any psychiatric disorder. The IQ scores of all participants with ASD were evaluated using either the WAIS-III or the WAIS-R, while those of typically developed individuals were estimated using the Japanese version of the national adult reading test (*Matsuoka et al., 2006*). Every participant with ASD was considered to be high functioning, because his or her full-scale IQ score was higher than 80. Participants completed the Japanese version of the autism-spectrum quotient (*Wakabayashi et al., 2007*). At the time of scanning, 25 ASD individuals were medication free, whereas the remaining 11 ASD individuals were administered the following psychotropic drugs: anxiolytics ($N = 4$), antidepressant ($N = 6$), antipsychotics ($N = 6$), anti-epileptics ($N = 2$), and sleep-inducing drugs ($N = 8$). Some participants were administered multiple drugs ($N = 7$). The Ethics Committee of the Faculty of Medicine of Showa University approved all the procedures used in this study, including the method of obtaining consent, in accordance with the Declaration of Helsinki. Written informed consent was obtained from all the participants after fully explaining the purpose of this study. Any concern regarding the possibility of reduced capacity to consent on his or her own was not voiced by either the ethics committee or patients' primary doctors.

### Working memory assessment

The SCZ patients and their healthy controls underwent the Japanese version of Brief Assessment of Cognition in Schizophrenia (BACS-J) (*Kaneda et al., 2007*). This cognitive battery is composed of six subtests including a digit sequencing test as a working memory measure. In this test, auditory sequences of numbers were presented, with increasing length from three to nine digits (*Figure 1B*). Participants repeated the sequences aloud by sorting in ascending order. The digit-sequencing scores were the number of correct trials among 28 trials ($18.4 \pm 4.1$ (SD), range 10 to 27 in patients while $22.9 \pm 4.3$ (SD), range 12 to 28 in controls). Their composite BACS score was evaluated by average score of BACS's five subtests other than the digit sequencing test.

### Examination of model prediction (SCZ patients and controls)

After preprocessing of fMRI data, FC was estimated following procedures described in the ATR dataset. We entered FC values into the normative model and predicted individual's working memory ability. For individual SCZ patients and controls, we performed a partial correlation analysis between the predicted and the actual working memory performance while factoring out age and composite BACS score excluding working memory (see above). We examined the statistical significance on the prediction accuracy by permutation tests as described in HCP dataset.

### Examination of model prediction (multiple diagnoses)

After prediction of working memory ability for each patient, we investigated working memory impairments in each of the four diagnoses. Specifically, patients' predicted letter 3-back working memory were evaluated by the Z-scores standardized to their age-and gender-matched controls collected in the same site. After confirming the homoscedasticity (Bartlett's test, p = 0.17), the standardized predicted working memory ability differences were entered in a one-way ANOVA with

diagnosis as a between-participant factor. Post-hoc pair-wise comparisons were corrected using Holm's method.

## Comparison of functional connectivity differences

We are interested in how working memory ability is determined by functional connectivity, and if the relationship between working memory ability and connectivity is altered by psychiatric disorders (e.g. if our model constructed from healthy controls can predict working memory of patients). The predicted working memory ability in our model is a weighted summation of connectivity values, meaning that alteration in working memory is determined by the product of connectivity values and model weights. For example, the working memory deficit caused by alteration of a specific connection is large, even if difference in a connectivity value between patients and controls is small, when the weight for the connection is large. Conversely, the working memory deficit is small, even if difference in a connectivity value is large, when the weight is small. Therefore, we mainly analyzed product of connectivity values and model weights.

To illustrate how each connection contributed to the predicted letter 3-back working memory ability differences, we defined the $D$-scores (difference-score) as follows. First, to align the FC value distribution of the control groups across the diagnostic groups, for every connection's Gaussian distribution $N(\mu, \sigma)$, each control group's FC value was transformed to an ATR dataset's distribution $N(\mu_{ATR}, \sigma_{ATR})$ using a linear transformation. The same transformation was performed for corresponding patient FC values. Since the predicted letter 3-back working memory ability is the weighted summation of the FC values, we can calculate each connection's differences in the weighted-FC values from the control average. Specifically, if $w_i$ is a regression weight and $x_{i,p}$ and $x_{i,c}$ are the FC values at connection $i$ of patient $p$ and control $c$, weighted-FC difference $D_{i,p}$ ($D$-score) becomes

$$D_i,p = w_i x_i,p - \mathrm{mean}(w_i x_i, c) = w_i\{x_i, p - \mathrm{mean}(x_i, c)\}.$$

To statistically compare the magnitude of the weighted-FC differences across diagnoses, the $D$-score was standardized for each patient and each connection:

$$Z_i,p = \{w_i x_i, p - \mathrm{mean}(w_i x_i, c)\}/\mathrm{SD}(w_i x_i, c) = D_i, p/\mathrm{SD}(w_i x_i, c).$$

Next, we examined the effects of diagnosis and connection on the $Z$-score. We tested the null hypotheses of (i) no main effect of diagnosis and (ii) no interaction effect between diagnosis and connection. Since these $Z$-scores showed heterogeneous variances across the diagnoses and connections, we calculated data-specific p values based on permutation tests as follows. First, to examine the main effect of diagnosis, we shuffled the diagnosis labels (i.e. SCZ, MDD, OCD, and ASD) and performed a two-way ANOVA with diagnosis as a between-participant factor and connection as a within-participant factor, obtaining the $F$ value for the main effect of diagnosis. We also performed post-hoc permutation tests to compare the disorder-pair-wise differences. We shuffled the diagnosis labels within a pair of diagnoses (e.g., SCZ and MDD), performed a two-way ANOVA, and obtained $F$ values for the main effect of diagnosis. Furthermore, we examined the interaction of diagnosis and connection by shuffling both the diagnosis and connection labels and performed a two-way ANOVA, obtaining $F$ values for interaction. These permutations were repeated 10,000 times for the main effects and 100,000 times for the interaction effect. The reported p values indicate how many times the observed $F$ values were obtained in the repetitions. A post-hoc Kruskal-Wallis test was performed to examine the simple main effects of the diagnosis on all the connectivity alteration $Z$-scores. We performed false discovery rate (FDR) correction to account for multiple comparisons (Benjamini-Hochberg method, $Q < 0.05$).

## Acknowledgements

This research was conducted as the 'Development of BMI Technologies for Clinical Application' of the Strategic Research Program for Brain Sciences supported by Japan Agency for Medical Research and Development (AMED). This research was supported by AMED under Grant Number JP18dm0307008. Drs. Yamashita, Kawato, and Imamizu were also supported by the ImPACT Program of Council for Science, Technology and Innovation (Cabinet Office, Government of Japan). Dr. Imamizu was also partially supported by JSPS KAKENHI Grant Number 26120002. Dr. Kasai was

partially supported by Brain/MINDS, AMED. Data were provided in part by the Human Connectome Project, WU-Minn Consortium (Principal Investigators: David Van Essen and Kamil Ugurbil; 1U54MH091657) funded by the 16 NIH Institutes and Centers that support the NIH Blueprint for Neuroscience Research; and by the McDonnell Center for Systems Neuroscience at Washington University.

## Additional information

### Funding

| Funder | Grant reference number | Author |
|---|---|---|
| Council for Science, Technology and Innovation | ImPACT Program | Masahiro Yamashita<br>Mitsuo Kawato<br>Hiroshi Imamizu |
| Japan Agency for Medical Research and Development | Brain/MINDS | Kiyoto Kasai |
| Wellcome Trust | | Ben Seymour |
| Arthritis Research UK | 21357 | Ben Seymour |
| Ministry of Education, Culture, Sports, Science, and Technology | 'Development of BMI Technologies for Clinical Application' of the Strategic Research Program for Brain Sciences and JP18dm0307008 | Mitsuo Kawato |
| Japan Society for the Promotion of Science | KAKENHI 26120002 | Hiroshi Imamizu |

The funders had no role in study design, data collection and interpretation, or the decision to submit the work for publication.

### Author contributions

Masahiro Yamashita, Conceptualization, Resources, Software, Formal analysis, Validation, Investigation, Visualization, Methodology, Writing—original draft, Project administration, Writing—review and editing; Yujiro Yoshihara, Ryuichiro Hashimoto, Noriaki Yahata, Naho Ichikawa, Yuki Sakai, Takashi Yamada, Noriko Matsukawa, Go Okada, Resources, Data curation; Saori C Tanaka, Kiyoto Kasai, Nobumasa Kato, Yasumasa Okamoto, Funding acquisition; Ben Seymour, Hidehiko Takahashi, Conceptualization, Supervision, Funding acquisition, Project administration, Writing—review and editing; Mitsuo Kawato, Conceptualization, Formal analysis, Supervision, Funding acquisition, Visualization, Methodology, Writing—original draft, Project administration, Writing—review and editing; Hiroshi Imamizu, Conceptualization, Supervision, Funding acquisition, Investigation, Methodology, Writing—original draft, Writing—review and editing

### Author ORCIDs

Masahiro Yamashita http://orcid.org/0000-0003-1520-2548
Ryuichiro Hashimoto http://orcid.org/0000-0002-9661-3412
Ben Seymour https://orcid.org/0000-0003-1724-5832
Hiroshi Imamizu http://orcid.org/0000-0003-1024-0051

### Ethics

Human subjects: ATR dataset was acquired using protocol (#12-101) according to the Declaration of Helsinki and approved by the Ethics Committee at Advanced Telecommunication Research Institute International. All participants gave written informed consent. Data from SCZ group was acquired by study design that was approved by the Committee on Medical Ethics (#R0027) of Kyoto University and was conducted in accordance with the Code of Ethics of the World Medical Association. All participants gave written informed consent. Data from MDD group was acquired by study protocol (#E-38) that was approved by the Ethics Committee of Hiroshima University. All participants gave written

informed consent. Data from OCD group was acquired by study protocol (#RBMR-C-1098-5) that was approved by the Medical Committee on Human Studies at the Kyoto Prefectural University of Medicine. All participants gave written informed consent. Data from ASD group at the University of Tokyo was acquired by study protocol (#3048 and #3150) approved by the Ethics Committee of the Graduate School of Medicine and Faculty of Medicine at the University of Tokyo. All participants gave written informed consent. Data from ASD group at Showa University was acquired by study protocol (#893) that was approved by Ethics Committee of the Faculty of Medicine of Showa University. All participants gave written informed consent.

## Decision letter and Author response
Decision letter https://doi.org/10.7554/eLife.38844.023
Author response https://doi.org/10.7554/eLife.38844.024

# Additional files

## Supplementary files
• Supplementary file 1. Resting state fMRI scan parameters and their values.
DOI: https://doi.org/10.7554/eLife.38844.017

• Supplementary file 2. Whole-brain intrinsic functional network labels and their component regions.
DOI: https://doi.org/10.7554/eLife.38844.018

• Transparent reporting form
DOI: https://doi.org/10.7554/eLife.38844.019

## Data availability
The following dataset was generated: Yamashita, M, Yoshihara, Y, Hashimoto, R, Yahata, N, Ichikawa, N, Sakai, Y,. .. Imamizu, H, 2018, Working Memory Prediction Database, https://bicr.atr.jp/dcn/en/download/database-wmp/. A download link for the open access dataset will be sent after the application form for data usage is completed (https://bicr.atr.jp/dcn/wp-content/uploads/Application_Form_for_Data_Usage_WMP-3.pdf). MATLAB code used to build the prediction model are also shared via this download link. You can send the completed application form to dcn_db@atr.jp. Part of this database includes data from the DecNef Project (https://bicr-resource.atr.jp/decnefpro/) and the project will share raw MRI data by the end of 2018. The Human Connectome Project 500 Subjects Release Open Access dataset is available from ConnectomeDB (https://db.humanconnectome.org/app/template/Login.vm) after the creation of a free account. Before accessing the dataset, users must agree with the Open Access Data Use Terms from ConnectomeDB (further information can be found here https://www.humanconnectome.org/study/hcp-young-adult/document/500-subjects-data-release and here https://www.humanconnectome.org/study/hcp-young-adult/data-use-terms).

The following dataset was generated:

| Author(s) | Year | Dataset title | Dataset URL | Database and Identifier |
|---|---|---|---|---|
| Masahiro Yamashita, Yujiro Yoshihara, Ryuichiro Hashimoto, Noriaki Yahata, Naho Ichikawa, Yuki Sakai, Takashi Yamada, Noriko Matsukawa, Go Okada, Saori C Tanaka, Kiyoto Kasai, Nobumasa Kato, Yasumasa Okamoto, Ben Seymour, Hidehiko Takahashi, Mitsuo Kawato, Hiroshi Imamizu | 2018 | Working Memory Pred | https://bicr.atr.jp/dcn/en/download/database-wmp/ | Working Memory Prediction Database, database-wmp |

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
