## [Decision Letter]

Thank you for submitting your article "A prediction model of working memory across health and psychiatric disease using whole-brain functional connectivity" for consideration by *eLife*. Your article has been reviewed by two peer reviewers, and the evaluation has been overseen by a Reviewing Editor and Michael Frank as the Senior Editor. The following individuals involved in review of your submission have agreed to reveal their identity: Monica Rosenberg (Reviewer #2); Xi-Nian Zuo (Reviewer #3).

The reviewers have discussed the reviews with one another and the Reviewing Editor has drafted this decision to help you prepare a revised submission.

Summary:

The central findings – transdiagnostic WM prediction – builds nicely on recent work on fingerprinting, prediction and transdiagnostic analyses. It is actually quite surprising that training a model on such a small set of healthy subjects generalizes across diagnoses and demographics that lie well outside the training data, but therein lies the value of the paper. While the relatively "dirty" acquisition and cohort details may not be ideal from a pure research perspective, it probably adds to the ecological validity and clinical translatability.

The paper has had two very thorough, excellent technical reviews. All major concerns are reasonable and should be addressed. Four concerns warrant specific commentary:

1) Reviewer 2's first point regarding the greater specificity of 2-back over 0-back to working memory and whether there are other cognitive processes at play.

2) Reviewer 3's first point regarding test-retest reliability. Ideally, you could pursue the reviewer's request here, although I note he has offered alternatives if this is not possible.

3) Much of the model (~34%) relies on left FP self-correlation – some sort of proxy for the internal coherence of that ICA map. None comes from the right FP – a slightly odd dependence on one feature and an asymmetry. It would be reassuring if this stood up to the test retest reliability analyses.

4) Given that you use parametric test statistics, it is not obvious why you also employed resampling to ascertain significance.

*Reviewer #2:*

In a training sample of 17 healthy adults, the authors built a model to predict d' on a 3-back task from between- and within-network resting-state functional connectivity. They applied the model to resting-state data from an independent sample of 474 healthy adults from the Human Connectome Project dataset and found that model predictions were significantly correlated with n-back task performance when controlling for fluid intelligence and motion, significantly (inversely) correlated with fluid intelligence when controlling for n-back performance and motion, and not significantly correlated with motion when controlling for n-back performance and fluid intelligence. They applied the model to a second independent sample of resting-state data from 58 individuals with schizophrenia and found that predictions were correlated with a working memory measure when controlling for general cognitive ability and age. Based on these external validation results they argue that the model is generalizable and specific to working memory abilities.

They next applied the model to three additional datasets with patient and control populations. They found that predicted degree of working memory impairment relative to matched controls was greatest for patients with schizophrenia followed by patients with major depressive disorder, obsessive compulsive disorder, and autism spectrum disorder. This ordering replicates the degree of working memory impairments reported by previous meta-analyses.

Overall this paper is a rigorous example of neuroimaging-based predictive modeling based on the generalization to two external validation datasets and between-group comparisons in three additional independent samples. My enthusiasm for the work is only slightly dampened by questions about the patient-vs.-control analyses, the specificity of the working memory model, and the feature importance analysis.

1) The authors take steps to show that model predictions are related to working memory ability specifically rather than cognitive ability more generally, but additional analyses would strengthen this claim. First, the measure of working memory in the HCP dataset includes 0-back task performance, which indexes sustained attention and attentional control rather than working memory. Are model predictions more closely related to 2-back than to 0-back task accuracy? Furthermore, because even 2-back and 3-back tasks measure a number of processes beyond working memory (Kane et al. 2007, J Exp Psychol Learn Mem Cogn), it would be informative to test whether model predictions are related to another measure of working memory such as performance on the NIH toolbox list-sorting task. Second, in the group-level analyses, are predicted working memory deficits more similar to working memory deficits observed in meta-analyses than they are to deficits observed in fluid intelligence or other cognitive domains (in terms of effect size or relative ordering across disorders)?

2) The lack of working memory measures presents challenges for the between-group comparisons in the patient samples. Although within each site the samples are matched on age and sex, the groups differ along a number of dimensions beyond working memory (e.g., medication status and potentially IQ and other cognitive abilities), and it's not clear whether patients and controls were scanned under the same protocols at each site, or whether protocols differed between sites. Limitations due to these potentially confounding factors should be clearly outlined in the manuscript. Related to this, are there working memory scores for the controls in the schizophrenia sample? If so, did predicted impairment reflect observed impairment in that sample, and does the model hold when applied to this full sample of patients and controls together?

3) It appears that the measure of general cognitive ability in the schizophrenia dataset includes a verbal memory measure. How correlated are working and verbal memory scores in this sample, and what is the justification for including it in the general cognitive ability score rather than treating it as a variable of interest?

4) Why was the HCP 500-subject release (2014) used rather than the 900-subject release (2015) or the 1200-subject release (2017)? Although the external generalization results are strong I would find it even more convincing if the model generalized to the full sample of HCP individuals.

5) The analysis of model weights and functional connectivity alterations between patient and control groups is somewhat confusing. Why does Figure 2 visualize the product of mean FC values and model weights (which will change depending on an individual's unique FC values) rather than just the raw model coefficients? Why do the patient/control difference scores incorporate model weights, rather than simply reflect changes in FC networks predicting working memory? It would be helpful to explain these choices in greater detail.

6) More details about the scrubbing procedure applied would be useful. Were frames before and after high-motion volumes excluded? What was the distribution of number of excluded volumes in each dataset? Did this differ by dataset or group?

7) The manuscript is lacking a discussion of predictive network anatomy, anatomy of networks that change vs. stay consistent between disorders, and implications for cognitive psychology or cognitive neuroscience. What do the current findings tell us about working memory and the functional networks that support it from a basic science perspective?

*Reviewer #3:*

The authors performed predictive models to examine FC-WMA across psychiatric diseases using a verbal 3-back task. This work was done using a set of data cohorts. The predicted effect size estimates on verbal WMA impairment were comparable to previous meta-analysis results. I personally enjoyed reading this manuscript. This is a very nice sample for reproducible brain-behavior association studies. I would be happy to support its acceptance of a publication in the *eLife* journal. However, I still have several concerns, which need to be fully addressed before the publication.

Summary of concerns:

1) It is highly important for studies using clinical patients to choose a measurement tool with high test-retest reliability. The authors employed ICA-derived networks as spatial profiles for whole brain FC modeling, however none of any references was given to support its reliability reaching to the clinically recognized request (ICC > 0.8). Is there any possibility of performing a test-retest analysis using public test-retest datasets (e.g., Consortium for Reliability and Reproducibility) to demonstrate the reliability matched to the level in clinic. At least, the literature on test-retest reliability of rfMRI metrics should be carefully documented if you cannot do it in the reasonable time frame (e.g., < 2months), see a review on this topic from my lab (PMID: 24875392). Meanwhile, dual regression with group ICA has been a highly reliable method, and the authors compared it with the FC method for the predictive modeling?

2) Head motion: Power et al. recently (PMID: 28880888) demonstrated that the order of performing preprocessing rfMRI data has effects on the performance of head motion removal. Of important relevance here is that the data should not be corrected for slice timing differences before the head motion estimated and reduced. The authors should check if their findings are influenced by such a change. Regarding the preprocessing, it is worth noting that ways of dealing with motion are different across data cohorts. How will this have an impact on reproducibility of the findings?

3) Demographical factors: It is widely known that age and sex have effects on FC, and how these two affect the observations reported here?

4) Figures: In Figure 1C, it is quite confusing that all the drawings of the graphical brain are the same across different clinical diagnoses (SCZ, MDD, OCD and ASD).

5) The authors have done a good work on dealing with head motion. However, just a curious point, several work also demonstrated potentially meaningful factors embedded in head motion as trait of human beings. At this point, interesting points related to the current work are: 1) Is there any relationship between motion and WMA? 2) Is there any correlation between global signal and WMA? 3) If so, what is the causal relationship among the four (motion, global signal, WMA and FC)?

6) Is there any plan in place to share the data publicly?

---

## [Author Response]

[…] The paper has had two very thorough, excellent technical reviews. All major concerns are reasonable and should be addressed. Four concerns warrant specific commentary:1) Reviewer 2's first point regarding the greater specificity of 2-back over 0-back to working memory and whether there are other cognitive processes at play.

To address this, we have performed additional analyses and – the details of which are in our response letter to reviewer #2. Briefly, the results can be summarized as follows: (i) we did not identify a significant difference in prediction accuracy of our model between the 2-back and 0-back scores; (ii) our model prediction was significantly correlated with another HCP working memory (list-sorting) score; (iii) the effect size of working memory deficits predicted by our model was more similar to digit span than general cognitive ability (IQ). Therefore, we demonstrated specificity of our model to working memory at two of three issues raised by reviewer #2.

2) Reviewer 3's first point regarding test-retest reliability. Ideally, you could pursue the reviewer's request here, although I note he has offered alternatives if this is not possible.

We elected to examine test-retest reliability of our methods for functional connectivity estimation that uses ICA-based intrinsic network definition. As suggested by reviewer #3, we used open data from the Consortium for Reliability and Reproducibility (CoRR) and calculated intra-class correlation (ICC) of functional connectivity values. These results are described in our response to reviewer #3’s comments, but briefly, we found that our methods are broadly comparable to other common connectivity estimation methods.

3) Much of the model (~34%) relies on left FP self-correlation – some sort of proxy for the internal coherence of that ICA map. None comes from the right FP – a slightly odd dependence on one feature and an asymmetry. It would be reassuring if this stood up to the test retest reliability analyses.

Our network definition is based on the BrainMap ICA which gives us useful information regarding the functional relevance of the ICA-derived networks based on meta-analyses of thousands of publications. In the BrainMap ICA paper (Laird et al., 2011; http://www.brainmap.org/icns/), IC18 (left fronto-parietal network; left FPN) showed greater functional relevance to working memory than IC15 (right FPN). Moreover, a meta-analysis on N-back task with different stimulus modality (Owen et al., 2005) found that monitoring of verbal stimuli was strongly associated with the left ventrolateral prefrontal cortex (a part of left FPN), while monitoring of spatial locations activated right lateralized frontal and parietal regions. In the current study, we used a letter 3-back task that requires encoding alphabet letters, which are more related to word monitoring than location monitoring. Therefore, the left FPN would be expected to contribute more to our prediction model than the right FPN.

Although much of the model relies on the within-network connectivity of the left FPN (~34% contribution), the right FPN also showed a substantial contribution to working memory via negative connectivity N2 (connection with the midbrain network, please see Figure 2) and N3 (connection with the superior parietal network) (~20% contribution).

We examined test-retest reliability by calculating intra-class correlation (ICC) for within-network connectivity of these networks using three different datasets from Consortium for Reliability and Reproducibility (CoRR) as fully described in our response to reviewer #3. Briefly, we found ICC values for the left FPN/right FPN: 0.28/0.23, 0.49/0.47, and 0.11/0.03 for BNU 1, IACAS 1, and Utah 1 dataset, respectively. According to an interpretation criteria of ICC (Landis and Koch, 1977), these results suggest that the two networks show comparable reliability with each other (0.2 < ICC < 0.5) for mid-term test-retest inter-session intervals (2-3 months: BNU 1 and IACAS datasets). Therefore, it is unlikely that difference in the test-retest reliability between the two networks induced the model’s dependence on left FPN.

We added the above issues to the Discussion section (eighth paragraph).

4) Given that you use parametric test statistics, it is not obvious why you also employed resampling to ascertain significance.

We applied the resampling tests to data shown in Figures 3A and 3C to ascertain significance of the correlation analysis between the predicted and actual working memory abilities in HCP datasets. However, we agree with the editor’s point that they are redundant. We have therefore removed the resampling results (Figures 3B and 3D) from our revised manuscript.

We used another resampling test in examination of effect of diagnosis and/or connection on degree of functional connectivity change (“*Z*-score”). Although a parametric two-way analysis of variance (ANOVA) requires homogenous variance across diagnosis/connection, some connections showed inhomogeneous variance across diagnoses. Thus, we used permutation tests to examine whether effect of diagnosis/connection significantly affects the degree of functional connectivity change. This is mentioned in the last paragraph of the subsection “Comparison of functional connectivity differences”.

Below, we provide an in-depth response to the individual issues raised by the reviewers.

Reviewer #2:[…] 1) The authors take steps to show that model predictions are related to working memory ability specifically rather than cognitive ability more generally, but additional analyses would strengthen this claim. First, the measure of working memory in the HCP dataset includes 0-back task performance, which indexes sustained attention and attentional control rather than working memory. Are model predictions more closely related to 2-back than to 0-back task accuracy? Furthermore, because even 2-back and 3-back tasks measure a number of processes beyond working memory (Kane et al. 2007, J Exp Psychol Learn Mem Cogn), it would be informative to test whether model predictions are related to another measure of working memory such as performance on the NIH toolbox list-sorting task. Second, in the group-level analyses, are predicted working memory deficits more similar to working memory deficits observed in meta-analyses than they are to deficits observed in fluid intelligence or other cognitive domains (in terms of effect size or relative ordering across disorders)?

First, we have examined whether the model prediction is more similar to the 2-back score than the 0-back score. Spearman’s rho partial correlation between the model prediction and task performance was 0.078 for 2-back task and 0.086 for 0-back task, while factoring out two confounding variables (fluid intelligence and head motion). There was no significant difference between the two correlation coefficients. Therefore, we could not conclude that the model prediction is more similar to 2-back score than 0-back score.

According the reviewer’s comment, we investigated the correlation between the predicted working memory ability and performance of a list-sorting task, which is a working memory task included in the NIH toolbox. The list-sorting score correlated positively with fluid intelligence (Spearman’s rho 0.32, *P* = 5.7 x 10^-13^) and negatively with head motion (Spearman’s rho -0.12, *P* = 0.009). Therefore, we performed a partial correlation analysis while factoring out these two variables. We found significant positive correlation between the predicted working memory ability and the list sorting score (Spearman’s rho = 0.084, *P* = 0.034). These results provide an evidence that the prediction model predicts working memory capability measured by another working memory task other than the 3-back task. These results and methods are described in the Results section (subsection “Prediction in Independent Test Set of Healthy Individuals”) and Materials and methods section (subsection “Human Connectome Project (HCP) Dataset”).

Second, we examined meta-analyses on fluid intelligence performed in the psychiatric diagnoses to examine if predicted working memory deficits are more similar to working memory deficits observed in meta-analyses than deficits in fluid intelligence. We found no systematic meta-analysis or review from a transdiagnostic viewpoint. Therefore, we searched for a meta-analysis on fluid intelligence for each diagnosis and found a small number of reports on this issue, as follows:

Schizophrenia: Heinrichs and Zakzanis reported that mean effect size is larger in full scale IQ measured by WAIS-Revised (*d* = -1.24, number of studies *k* = 35) than digit span (*d* = -0.61, *k* = 18). Rajji et al. reported that effect size is larger in full scale IQ (first-episode: *d* = 0.89, *k* = 29; youth-onset: *d* = -1.77, *k* = 15; late-onset: *d* = -1.61, *k* = 4) than digit span (first-episode: *d* = 0.64, *k* = 24; youth-onset: *d* = -0.85, *k* = 7; late-onset: *d* = -0.87, *k* = 5). These two meta-analyses commonly suggest that schizophrenia patients show larger effect size in IQ reduction than working memory deficits.

MDD: We found a review paper about fluid intelligence in first episode depression. Ahern and Semkovska reported that mean effect size is larger in digit span (forward: *d* = -0.35, *k* = 3; backward: *d* = -0.33, *k* = 4) than IQ composite (*d* = -0.26, *k* = 10). The IQ composite effect size is similar to predicted working memory effect size (*d* = -0.29).

OCD: We found a meta-analysis on fluid intelligence in OCD patients. Abramovitch et al. reported that mean effect size in full scale IQ in OCD was -0.35 (*k* = 40). Snyder et al. reported smaller effect size for digit span forward (*d* = -0.08, *k* = 19) and backward (*d* = -0.21, *k* = 11).

ASD: We found no meta-analysis regarding general cognitive ability or intelligence.

Figure 4—figure supplement 1 shows effect sizes of deficit for fluid intelligence (IQ) as reported in the above meta-analyses (with the largest *k* for each diagnosis) in comparison to those for working memory. Note that the predicted working memory ability falls within the confidence interval of the IQ effect size only for first-episode MDD while it falls within confidence interval of effect size of working memory (forward digit span) for every diagnosis. Regarding the relative order of effect sizes, the effect size of fluid intelligence deficits can be ordered as schizophrenia, OCD, and MDD (first episode). In contrast, the effect sizes of working-memory deficits (as measured by digit-span task) can be ordered as schizophrenia, MDD and OCD, which is consistent with the order predicted by our model. Therefore, predicted working-memory deficits were more similar to observed deficits in working memory than those in fluid intelligence. These results are described in the Results section (subsection “Prediction in Four Distinct Psychiatric Disorders”, last paragraph) of our revised manuscript.

Collectively, although we cannot conclude that the model prediction is more similar to 2-back score than 0-back score, the model was able to predict another measure of working memory (i.e. List-sorting). Moreover, predicted working memory deficits are more similar to working-memory deficits observed in meta-analyses than deficits in fluid intelligence.

2) The lack of working memory measures presents challenges for the between-group comparisons in the patient samples. Although within each site the samples are matched on age and sex, the groups differ along a number of dimensions beyond working memory (e.g., medication status and potentially IQ and other cognitive abilities), and it's not clear whether patients and controls were scanned under the same protocols at each site, or whether protocols differed between sites. Limitations due to these potentially confounding factors should be clearly outlined in the manuscript. Related to this, are there working memory scores for the controls in the schizophrenia sample? If so, did predicted impairment reflect observed impairment in that sample, and does the model hold when applied to this full sample of patients and controls together?

We agree with the reviewer and added limitations to the Discussion section as follows:

“Second, working memory performance was measured only in schizophrenia patients but not in other diagnostic groups. Therefore, it was impossible to compare their predicted working memory ability with measured scores. This presents challenges for the between-group comparisons in the patient samples.”

“Third, although the participants are matched on age and sex within each site, the groups may differ along a number of dimensions beyond working memory (e.g., medication status, scanning protocol, and potentially IQ and other cognitive abilities). It is difficult to fully control every dimension, and little is known how such dimensions affect estimation of functional connectivity.”

We added lines to the table in Supplementary file 1 to make it clear how many patients and controls were scanned with each scanner and protocol.

We newly analyzed data including the full sample of schizophrenia patients and controls, and found a consistent result with the previous result in only the patient samples. Specifically, we found that the digit-sequencing score was correlated positively with composite BACS score excluding working memory (see our reply to the next comment) (*r* = 0.68, *P* = 2.0 x 10^-17^), and negatively with age (*r* = -0.36, *P* = 5.7 x 10^-5^), but not with head motion (*r* = -0.04, *P* = 0.68). While controlling the age and the composite BACS score, a partial correlation analysis showed that the model prediction is significantly correlated with the digit-sequencing score (*ρ* = 0.15, *P* = 0.048).

These results are included in the Results section of the revised manuscript (subsection “Prediction in Individual Schizophrenia Patients and Controls”, last paragraph).

3) It appears that the measure of general cognitive ability in the schizophrenia dataset includes a verbal memory measure. How correlated are working and verbal memory scores in this sample, and what is the justification for including it in the general cognitive ability score rather than treating it as a variable of interest?

First, we found positive correlation between observed working memory and verbal memory scores of BACS in the schizophrenia patients (*r* = 0.49, *P* = 1.0 x 10^-3^). Moreover, the working memory scores showed positive correlation with the other four sub scores (‘verbal fluency’, ‘motor speed’, ‘executive function’, and ‘attention and processing speed’) (see Author response image 1). Therefore, we needed to factor out effects of these correlated scores to examine if the prediction model was specific to the working memory.

**Author response image 1. respfig1:** Pearson’s correlation matrix of BACS sub-scores.

The reason why we treated verbal memory as a variable of no-interest is that the verbal memory measured by BACS seemed very different from working memory measured by the letter 3-back task. We constructed the prediction model based on the letter 3-back task. This task is a recognition test in which alphabet letters (e.g. B, J) are presented moment to moment, and requires participants to update their memory and to group the letters into a unit (chunking). In contrast, the BACS verbal memory task is a free recall test of 15 words, so participants are required to recall lexical/semantic representations. Therefore, we assumed that cognitive functions necessary for performing the 3-back task are different from functions necessary for performing the verbal memory.

Thanks to reviewer #2’s comments, we realized that “verbal 3-back task” was a confusing term and changed it to “letter 3-back task”. Also, we found that the word “general cognitive ability” is not appropriate. Now we call it “composite BACS score excluding working memory”.

4) Why was the HCP 500-subject release (2014) used rather than the 900-subject release (2015) or the 1200-subject release (2017)? Although the external generalization results are strong I would find it even more convincing if the model generalized to the full sample of HCP individuals.

When we analyzed the data, only HCP 500-subject dataset was released. We are eager to test generalization of our model to the full sample of HCP. However, our analysis needs calculation of voxel-wise correlation when we calculate functional connectivity within a network (e.g., left fronto-parietal network). This calculation is time consuming process, which needs 8 hours for each subject. Therefore, it was impossible for us to finish the analysis of the full sample of HCP within a reasonable (i.e. two month) time-frame.

5) The analysis of model weights and functional connectivity alterations between patient and control groups is somewhat confusing. Why does Figure 2 visualize the product of mean FC values and model weights (which will change depending on an individual's unique FC values) rather than just the raw model coefficients? Why do the patient/control difference scores incorporate model weights, rather than simply reflect changes in FC networks predicting working memory? It would be helpful to explain these choices in greater detail.

We are interested in how working memory ability is determined by functional connectivity, and if the relationship between working memory ability and connectivity is altered by psychiatric disorders (e.g., if our model constructed from healthy controls can predict working memory of patients). The predicted working memory ability in our model is a weighted summation of connectivity values, meaning that alteration in working memory is determined by the product of connectivity values and model weights. For example, the working memory deficit caused by alteration of a specific connection is large, even if difference in a connectivity value between patients and controls is small, when the weight for the connection is large. Conversely, the working memory deficit is small, even if difference in a connectivity value is large, when the weight is small. Therefore, we mainly analyzed product of connectivity values and model weights.

We added the above explanations to our revised manuscript’s Methods and Materials section (subsection “Comparison of functional connectivity differences”, first paragraph).

6) More details about the scrubbing procedure applied would be useful. Were frames before and after high-motion volumes excluded? What was the distribution of number of excluded volumes in each dataset? Did this differ by dataset or group?

Regarding the first question, frames were excluded if the motion was excessive at each time-point (frame-wise displacement > 0.5 mm). We did not remove a frame before or after the excessive motion. We added these details about the scrubbing procedure to the Materials and methods section:

“We performed slice timing correction and then motion estimation. The estimated motion parameters were used to estimate excessive motion data by frame-wise displacement > 0.5 mm. We did not remove a frame before or after the excessive motion”.

We calculated the ratio of excluded volumes to the total number of volumes for each subject, and averaged within patients or controls for each diagnosis. They were 2.3 ± 5.5% / 1.4 ± 2.9% (patients/controls) for SCZ, 2.7 ± 6.6% / 2.4 ± 6.3% for MDD, 0.4 ± 0.8% / 0.7 ± 1.7% for OCD, and 1.4 ± 3.8% / 4.6 ± 8.5% for ASD. We found a significant difference in the ratio between patients and controls only for ASD (*t*_97.3_= 2.91, *P* = 4.4 x 10^-3^). However, it is unlikely that this difference caused a problem in our result because there was not a significant difference in the predicted working memory ability between ASD patients and their controls. We reported the ratios for each diagnosis (subsection “Multiple Psychiatric Diagnoses Dataset”, first paragraph).

7) The manuscript is lacking a discussion of predictive network anatomy, anatomy of networks that change vs. stay consistent between disorders, and implications for cognitive psychology or cognitive neuroscience. What do the current findings tell us about working memory and the functional networks that support it from a basic science perspective?

To increase interpretability of our results from the perspective of cognitive neuroscience, we grouped the 18 networks into seven clusters, each of which is more applicable to a functional understanding. We investigated how alterations of connections between these clusters affected working memory ability. We added the results (Figure 5—figure supplement 2) to the Results section and our interpretations to the Discussion section as follows:

In the Results section:

“To understand these results from global brain networks, we grouped the 18 networks into seven clusters based on the hierarchical clustering of the networks performed in the BrainMap ICA study (Laird et al., 2011). […] Note that we could not find any connections that have a node in the default-mode cluster, which did not appear in the figure.”

In the Discussion section:

“Our results identified alterations in large-scale network clusters that correlated with working memory impairment (Figure 5—figure supplement 2). […] This is consistent with our result that alterations of connections related to the motor/visuospatial networks were associated with lower working memory ability in schizophrenia, MDD, and ASD.”

Reviewer #3:[…] 1) It is highly important for studies using clinical patients to choose a measurement tool with high test-retest reliability. The authors employed ICA-derived networks as spatial profiles for whole brain FC modeling, however none of any references was given to support its reliability reaching to the clinically recognized request (ICC > 0.8). Is there any possibility of performing a test-retest analysis using public test-retest datasets (e.g., Consortium for Reliability and Reproducibility) to demonstrate the reliability matched to the level in clinic. At least, the literature on test-retest reliability of rfMRI metrics should be carefully documented if you cannot do it in the reasonable time frame (e.g., < 2months), see a review on this topic from my lab (PMID: 24875392). Meanwhile, dual regression with group ICA has been a highly reliable method, and the authors compared it with the FC method for the predictive modeling?

We thank the reviewer for pointing out the important measurement of test-retest reliability. As the reviewer suggested, we examined ICC for the resting state functional connectivity method using Consortium for Reliability and Reproducibility (CoRR) database. We added the text below to the Results, Discussion, and Materials and Methods sections respectively.

as follows.

Materials and methods:

“1. Datasets

We analyzed data from Consortium for Reliability and Reproducibility (CoRR) that facilitates assessment of test-retest reliability and reproducibility for resting state functional connectivity (Zuo et al., 2014). We picked up following three datasets, Beijing Normal University (BNU 1), Institute of Automation, Chinese Academy of Sciences (IACAS 1), and University of Utah (Utah 1). We selected these datasets because 1) they have test-retest data across fMRI sessions, 2) ages of participants are comparable with those in our discovery dataset that was used for the construction of our model (ATR dataset), 3) two datasets include Asian participants (participants in ATR dataset are Japanese), 4) sizes of datasets are relatively small (we had to finish our analysis within 2 months).

[…]

3. Test–retest reliability

Intra-class correlation (ICC) was calculated for each of the 171 functional connectivity values (univariate test-retest reliability). ICC was calculated by following equation:

ICC = (MS_b_ – MS_w_)/{MS_b_ + (*k –* 1)MS_w_}

where, MS_b_ is the between-subjects mean squared error and MS_w_ is the within-subjects mean squared error and k is the number of independent fMRI measures (i.e., *k* = 2 for test and retest). We put negative ICC values to be zeros as done by previous studies (e.g., Zhang et al., 2011).”

Results:

“We obtained ICC values 0.34 ± 0.12 (range 0 to 0.65) for BNU 1, 0.26 ± 0.18 (range 0 to 0.66) for IACAS 1, and 0.21 ± 0.17 (range 0 to 0.59) for Utah 1 datasets. We found ICC values for the left FPN/right FPN: 0.28/0.23, 0.49/0.47, and 0.11/0.03 for BNU 1, IACAS 1, and Utah 1 dataset, respectively.”

Discussion

“According to an interpretation criteria of ICC (Landis and Koch, 1977), our connectivity estimation methods yielded “fair” reliability (0.2 < ICC ≤ 0.4) for the three datasets. A previous study on test-retest reliability of functional connectivity between 18 different brain regions (Birn et al., 2013), reported similar ICC values (ICC ~ 0.2) when scan length was 6 to 15 minutes. Another previous study examined functional connectivity reliability (Noble et al., 2017), using 268 regions from whole-brain also reported that 6 min of scan length yielded similar reliability (dependability coefficient ~ 0.2 to 0.4). Although our connectivity estimation methods cannot reach clinically recognized request (ICC > 0.8), these studies suggest that test-retest reliability of our methods are comparable to other common connectivity estimation methods.”

2) Head motion: Power et al. recently (PMID: 28880888) demonstrated that the order of performing preprocessing rfMRI data has effects on the performance of head motion removal. Of important relevance here is that the data should not be corrected for slice timing differences before the head motion estimated and reduced. The authors should check if their findings are influenced by such a change. Regarding the preprocessing, it is worth noting that ways of dealing with motion are different across data cohorts. How will this have an impact on reproducibility of the findings?

We re-analyzed the data by conducting the motion estimation before slice timing correction. We summarize the results below:

Regarding the individual-level prediction on the SCZ dataset, we found a marginally significant positive correlation between the predicted working memory and the digit-sequencing score (*r* = 0.21, *P* = 0.059), but partial correlation analysis did not show a significant correlation (*rho* = 0.10, *P* = 0.26) while factoring out the BACS composite score and age. We found the model prediction correlated not only with the digit-sequencing score but also with the BACS composite score (*r* = 0.21, *P* = 0.056). Based on the strong correlation between the digit-sequencing and the BACS composite score (*r* = 0.61), obtaining prediction specific to the subtest was a quite challenging goal. These results may reduce the model’s specificity to working memory at least for SCZ dataset but still the model prediction was related to cognitive ability rather than other confounding factors (age and motion).

Regarding the multiple psychiatric diagnoses dataset, we found almost the same results as our previous results even after changing this procedure, as follows.

We found two additional MDD patients with excessive motion (40% of data showed frame-wise displacement > 0.5 mm) and removed them from further analysis. We detected outliers in the model prediction within each group (defined as values > 3 SD from the mean): a patient of SCZ, two control participants of MDD, and a patient with OCD.

We identified significant differences in the predicted working memory between the patient and controls only for SCZ patients (two-tailed *t*-test for SCZ group: *t*_115_ = -3.11, *P* = (2.4 x 10^-3^) x 4 = 0.0096, Bonferroni corrected). This result is similar to the original results:

Original manuscript: “We identified significant differences in the predicted working memory between the patient and controls only for SCZ patients (two-tailed *t*-test for SCZ group: *t*_116_ = -3.68, *P* = (3.5 x 10^-4^) x 4 = 0.0014, Bonferroni corrected; Figure 4A)”

Next, we calculated individual patients’ *Z*-score (normalized difference between a patient and average of controls at the same site) of the predicted working memory for each diagnosis. A one-way ANOVA revealed a significant main effect of diagnosis on the *Z*-score (*F*_3,242_ = 6.09, *P* = 5.2 x 10^-4^). The severity of the predicted impairment in SCZ patients was larger than all other diagnoses (post-hoc Holm’s controlled *t*-test, adjusted *P* < 0.05). These results are similar with the original results:

Original manuscript: “A one-way ANOVA revealed a significant main effect of diagnosis on the *Z*-score (*F*_3,245_ = 7.63, *P* = 6.8 x 10^-5^). The severity of the predicted impairment in SCZ patients was larger than all other diagnoses (post-hoc Holm’s controlled *t*-test, adjusted *P* < 0.05).”

The predicted working memory alteration was more negative in the order of SCZ, MDD, OCD, and ASD with effect sizes (Hedge’s g) of -0.57, -0.29, -0.18, and 0.15, respectively. These results are similar to the original results:

Original manuscript: “The predicted working memory alteration was more negative in the order of SCZ, MDD, OCD, and ASD with effect sizes (Hedge’s *g*) of -0.68, -0.29, -0.16, and 0.09, respectively.”

These results of the group-level analyses were almost the same as our previous results after changing the order of slice timing correction and motion estimation.

3) Demographical factors: It is widely known that age and sex have effects on FC, and how these two affect the observations reported here?

At the model building stage, we found no significant effect of age and sex on predicted working memory as described in our revised manuscript:

“We did not find a significant correlation between the predicted letter 3-back learning performance and age (*r* = 0.21, *P* = 0.42), gender (*r* = 0.28, *P* = 0.28)”.

Next, we examined if age affected predicted working memory in multiple psychiatric diagnoses dataset. We found a significant negative correlation between age and predicted working memory in ASD at site 1 (*r* = -0.38, *P* = 1.8 x 10^-3^), suggesting that younger participants were predicted to have greater working memory. We found no significant effect of age on predicted working memory in the other groups.

We also examined how predicted working memory differed between males and females within each diagnosis. Consequently, we found no significant effect of sex on predicted working memory (*t*-tests: SCZ *t*_116_ = 0.37, *P* = 0.72; MDD *t*_137_ = 1.13, *P* = 0.26, OCD *t*_90_ = 0.17, *P* = 0.86, ASD *t*_138_ = 1.28, *P* = 0.20).

Because age and sex were controlled between patients and controls in each diagnosis (Table 1), it is unlikely that age and sex affect the observations in this study.

4) Figures: In Figure 1C, it is quite confusing that all the drawings of the graphical brain are the same across different clinical diagnoses (SCZ, MDD, OCD and ASD).

We thank the reviewer for pointing this out. We have changed the thickness of the lines for each diagnoses that illustrate connectivity difference among different diagnoses.

5) The authors have done a good work on dealing with head motion. However, just a curious point, several work also demonstrated potentially meaningful factors embedded in head motion as trait of human beings. At this point, interesting points related to the current work are: 1) Is there any relationship between motion and WMA? 2) Is there any correlation between global signal and WMA? 3) If so, what is the causal relationship among the four (motion, global signal, WMA and FC)?

1) We examined correlation between head motion and working memory.

ATR dataset: There was no significant correlation between head motion and observed 3-back task performance (*r* = 0.23, *P* = 0.37).

SCZ dataset:

Patients only (*N* = 58), there was no significant correlation between head motion and observed digit-sequencing performance (*r* = -0.03, *P* = 0.83).

Patients and controls (*N* = 118), there was no significant correlation between head motion and observed digit-sequencing performance (*r* = -0.04, *P* = 0.68).

HCP dataset: There was a significant correlation between head motion and observed N-back accuracy (Spearman’s rank correlation *rho* = -0.24, *P* = 1.5 x 10^-7^) and list-sorting score (Spearman’s rank correlation = -0.12, *P* = 0.009). Note that we factored out the head motion using a partial correlation analysis when we investigated a correlation between the predicted and actual working memory performance.

2) and 3) To our understanding, for calculation of global signal, we need to subtract the fMRI signal in a baseline period from that in a task period for every voxel in the brain and then average the subtracted values across the voxels. Such subtraction is needed because an fMRI signal value is arbitrary one which changes according to runs. However, in the current study, we measured only resting-state fMRI in which there was no baseline or task period. Therefore, we were unable to address the questions about the global signal.

6) Is there any plan in place to share the data publicly?

We shared data in which informed consents for data sharing were obtained from participants at https://bicr.atr.jp/dcn/en/download/database-wmp/. We clarified which data are not allowed to be shared due to the lack of informed consents in Supplementary file 1.

ATR dataset:

https://bicr.atr.jp/dcn/en/download/database-wmp/

Multiple psychiatric diagnoses dataset:

https://bicr.atr.jp/dcn/en/download/database-wmp/

HCP dataset:

https://www.humanconnectome.org

These are specified in “Data availability” subsection in our revised manuscript.